# Oxidative phosphorylation is a key feature of neonatal monocyte immunometabolism promoting myeloid differentiation after birth

Greta Ehlers[1,14], Annika Marie Tödtmann [1,14], Lisa Holsten [2,3,4,5,14], Maike Willers [1], Julia Heckmann[2], Jennifer Schöning[2], Maximilian Richter[2], Anna Sophie Heinemann[1], Sabine Pirr [1], Alexander Heinz[6], Christian Dopfer[1], Kristian Händler [4,7], Matthias Becker [4,8], Johanna Büchel[9], Achim Wöckel[9], Constantin von Kaisenberg [10], Gesine Hansen[1,11], Karsten Hiller [6], Joachim L. Schultze [3,4,5], Christoph Härtel[2], Wolfgang Kastenmüller [12], Martin Vaeth [12], Thomas Ulas[3,4,5] & Dorothee Viemann [1,2,11,13] ✉

Neonates primarily rely on innate immune defense, yet their inflammatory responses are usually restricted compared to adults. This is controversially interpreted as a sign of immaturity or essential programming, increasing or decreasing the risk of sepsis, respectively. Here, combined transcriptomic, metabolic, and immunological studies in monocytes of healthy individuals reveal an inverse ontogenetic shift in metabolic pathway activities with increasing age. Neonatal monocytes are characterized by enhanced oxidative phosphorylation supporting ongoing myeloid differentiation. This phenotype is gradually replaced during early childhood by increasing glycolytic activity fueling the inflammatory responsiveness. Microbial stimulation shifts neonatal monocytes to an adult-like metabolism, whereas ketogenic diet in adults mimicking neonatal ketosis cannot revive a neonate-like metabolism. Our findings disclose hallmarks of innate immunometabolism during healthy postnatal immune adaptation and suggest that premature activation of glycolysis in neonates might increase their risk of sepsis by impairing myeloid differentiation and promoting hyperinflammation.

The immune response of newborn infants and adults to microbial challenges differs considerably[1,2], pointing to profound reprogramming of immunity during childhood. However, our knowledge about the molecular mechanisms that drive the age-dependent immunological differences and reprogramming during childhood is still fragmentary.

While lacking a fully developed adaptive immune system, newborns primarily rely on innate immune responses for protection against pathogens[1,2]. The capacity of neonatal (NEO) innate immune cells to mount inflammatory responses to a broad range of microbial stimuli is typically restricted compared to adults. This peculiarity of neonatal immune cells is often considered as a sign of immaturity and utilized to explain severe infections and sepsis in neonates. Conversely, it is increasingly acknowledged as physiological tolerance preventing exceeding inflammatory responses to microbial challenges in the new environment after birth[1,3–5].

The programming of energy metabolism is one of the most important determinants of an immune phenotype as a switch from mitochondrial oxidative phosphorylation (OXPHOS) to glycolysis usually provides the rapid energy supply that is required for strong immune responses to microbial challenges[6,7]. However, our knowledge about the metabolic programming of NEO innate immune cells has considerable gaps. Metabolic cell studies using peripheral blood from newborns or infants are lacking. Hitherto, only cord blood monocytes (Mo) from preterm and term newborns were studied showing in both of them reduced basal glycolytic activity compared to adult (AD) Mo. This could be associated with an impaired inflammatory response to *Candida* and lipopolysaccharide (LPS), however only in preterm but not in term Mo[8]. Another study generated cord blood-derived polarized macrophages and observed reduced glycolytic as well as reduced OXPHOS activity compared to AD blood-derived macrophages, but did not link this to immune phenotypes directly[9]. Thus, it remains elusive how the metabolic programming of innate immune cells in neonates determines their antimicrobial immune responses, as well as which cellular metabolic programming is advantageous at what age to meet the demands of age-specific cell functions.

In this work, we show how the energy metabolism and the activity of metabolic pathways change over age in primary human Mo collected from healthy newborns (cord blood), infants (1 month to 12 months old), toddlers and preschoolers (2 to 5 years old), and adults. By using transcriptomic, metabolic, and immunological approaches, we reveal age-dependent metabolic Mo profiles that are linked to age-specific Mo functions. Our findings show that the metabolism of Mo follows a unidirectional reprogramming after birth, which is important to consider when tailoring age-appropriate intervention strategies that modulate the cell metabolism.

## Results

### The differential response of AD and NEO Mo to LPS stimulation is linked to an age-specific baseline programming of metabolism

To decipher the age-specific transcriptional programming of Mo, we performed RNA sequencing of untreated and LPS-activated Mo isolated from cord blood of healthy newborn infants and peripheral blood of healthy adults (Fig. 1a). Principal component analysis (PCA) (Fig. 1b) and the number of differentially expressed genes (DEGs) (Supplementary Fig. 1a, b, and Supplementary Data 1) showed that LPS stimulation induced stronger transcriptional changes in AD compared to NEO Mo, though 82% of the LPS-induced genes in NEO Mo overlapped with those upregulated in AD Mo (931 shared of 1135 upregulated DEGs in NEO LPS vs Ctrl) (Supplementary Fig. 1c). Interestingly, the overlap of genes downregulated by LPS in NEO Mo and AD Mo was only 5% (51 shared of 1037 downregulated DEGs in NEO LPS vs Ctrl) (Supplementary Fig. 1d). The PCA revealed a clear separation by age on the second principal component, as well as a nearly parallel shift upon LPS stimulation along the first principal component, suggesting that the age-specific baseline programming defines the differential LPS responsiveness. However, the total overlap between DEGs in NEO *versus* AD Mo after LPS stimulation and DEGs in NEO *versus* AD at baseline was only 16% (180 shared of 531 upregulated genes after LPS stimulation; 70 shared of 1062 downregulated genes after LPS stimulation) (Supplementary Fig. 1c, d), indicating that it is not a simple recapitulation of age-dependent differential gene expression in basal and LPS-activated states.

To identify the age-dependent transcriptional differences at baseline that link to the differential LPS response, we performed gene co-expression network analysis that groups genes based on expression profiles into similarly regulated (color-coded) gene modules (Fig. 1c, Supplementary Fig. 1e), allowing detection of shared and distinct transcriptional signatures between NEO and AD Mo at baseline and after LPS stimulation (Supplementary Fig. 1f). Differential expression after LPS stimulation was strongest in genes belonging to the modules

seagreen and lightgreen with strong upregulation in AD Mo but only minor LPS-induction in NEO Mo (Fig. 1c, Supplementary Fig. 1e, f, Supplementary Data 1). Functional enrichment based on gene ontology (GO), Hallmark (HM) and KEGG (KG) annotations associated these genes with inflammatory immune response terms such as "TNFA signaling via NFKB" (Fig. 1d, Supplementary Data 1). Higher LPS-induced TNF production by AD Mo compared to NEO could also be validated at the protein level (Fig. 1e).

At baseline, the strongest gene expression differences were detected in the modules maroon and steelblue. Genes of the maroon module were lower expressed in AD compared to NEO, in fact both at baseline and after LPS activation, and enriched for cell cycle control and chromatin remodeling functions. Their increased expression might be related to more active myeloid differentiation processes in NEO Mo, as NEO Mo distinguish from AD Mo by a higher GM-CSF production (Fig. 1f). Oppositely, genes of the steelblue module showed higher basal expression in AD than in NEO Mo and were thereby linked to the similarly differential expression (higher in AD than NEO Mo) of inflammatory genes (seagreen and lightgreen modules) after LPS activation. Functional enrichment highlighted an association of the steelblue module genes with glycolysis, suggesting higher basal glycolytic activity in AD than NEO Mo.

To assess how NEO and AD Mo use glucose, we performed U-[13 C]-glucose tracing studies (Fig. 1g). We found comparable isotopic enrichment in m3+ isotopologues of lactate but elevated m3+ pyruvate (Fig. 1h), supporting enhanced glycolysis in AD Mo compared to NEO Mo. In contrast, isotopic enrichment in tricarboxylic acid cycle (TCA) metabolites pointed to an increased TCA and OXPHOS activity in NEO Mo compared to AD Mo (Fig. 1i). Higher levels of m2+ citrate in NEO Mo indicated enhanced pyruvate metabolism and intermediate transfer to the TCA, while lower m2+ succinate levels with slightly increased m2+ malate levels ($p = 0.0676$) reflected higher electron transfer and OXPHOS activity[6,10].

Taken together, these data suggested that the lower inflammatory response of NEO Mo to LPS is linked to the lower baseline glycolytic activity, while this might be inversely associated with vivid cell differentiation processes that come with high OXPHOS activity.

### Metabolic reprogramming during the first years of life shifts Mo from OXPHOS to glycolytic dependence

To elucidate how the metabolic state of Mo changes during the first years of life, we applied SCENITH to blood sampled from healthy newborns, infants, toddlers, and adults (Fig. 2a, Supplementary Table 1). SCENITH allows quantifying protein translation by puromycin incorporation as a measure of their global metabolic activity and its dependence on OXPHOS (mitochondrial dependence, MD) and glycolysis (glucose dependence, GD) (Fig. 2b)[11]. NEO Mo were characterized by high metabolic activity (Fig. 2c) with a high dependence on OXPHOS (Fig. 2d) but low dependence on glycolysis (Fig. 2e), supporting the isotope tracing studies (Fig. 1h, i). With increasing age, this metabolic phenotype shifted towards lower metabolic activity with high dependence on glycolysis but low dependence on OXPHOS in AD Mo. The overall relationship between OXPHOS and glycolytic dependence in Mo of all ages was inverse (Fig. 2f). Correlations with the cytokine markers of inflammatory responsiveness and cell differentiation revealed that a high glycolytic dependence came with an increased TNF response (Fig. 2g), whereas a high OXPHOS dependence was associated with high GM-CSF levels (Fig. 2j). In contrast, there was no correlation between OXPHOS dependence and TNF levels (Fig. 2i) or glycolytic dependence and GM-CSF levels (Fig. 2h). These data, combined with the transcriptomic and isotope tracing data, corroborated that NEO Mo are characterized by ongoing cell differentiation that is linked to high OXPHOS activity at the cost of glycolytic activity and inflammatory responsiveness.

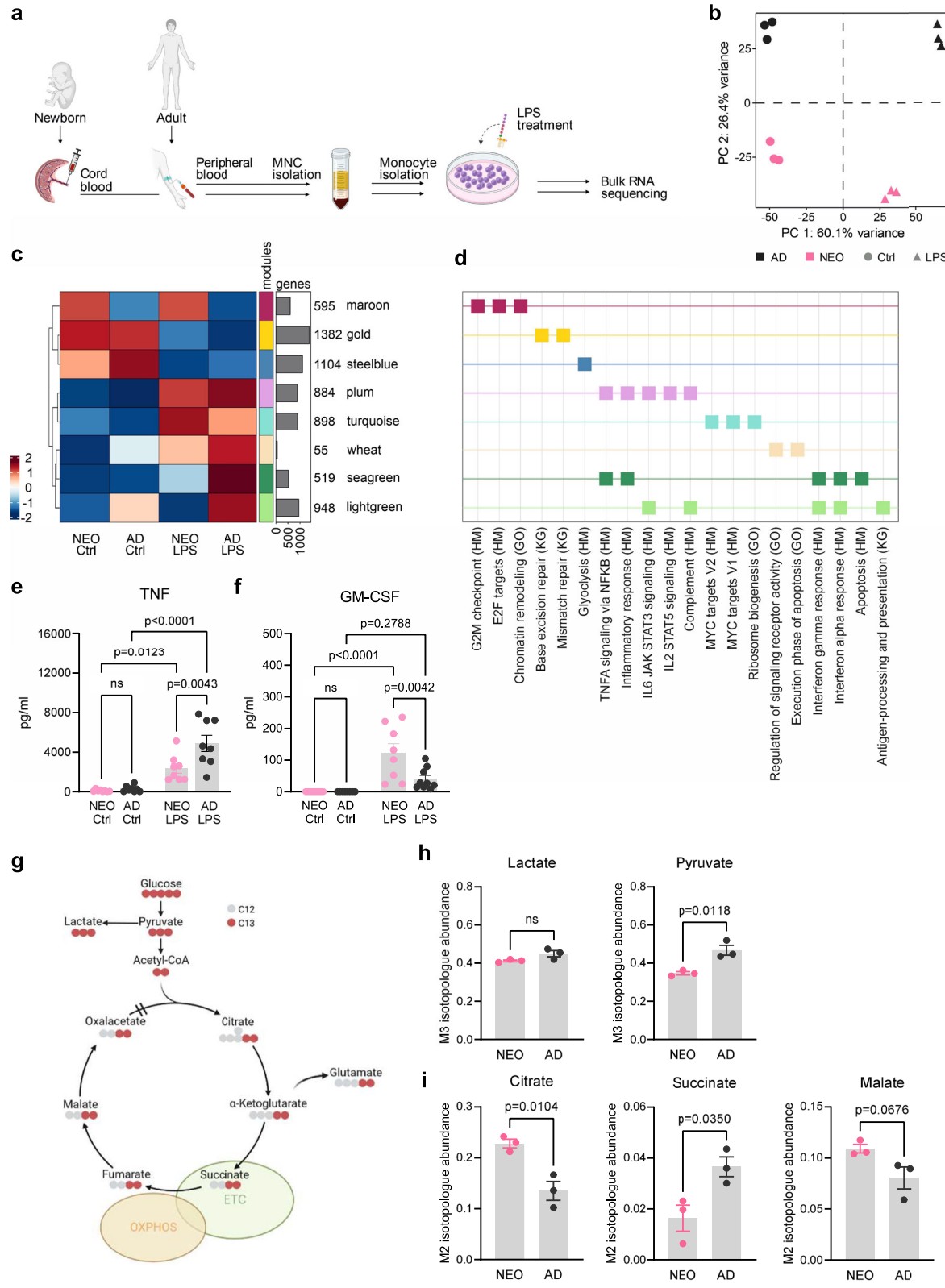

To determine the potential effect of endogenous and environmental factors on the metabolic phenotype of Mo, we built generalized linear models to integrate the metabolic parameters with clinical parameters. Age was the strongest positively influencing factor on energy metabolism (Supplementary Fig. 2a) and OXPHOS (Fig. 2k) and the strongest negatively influencing factor on glycolysis (Supplementary Fig. 2b). In contrast, sex, gestational age (GA), mode of delivery (MOD), body weight, and the number of vaccinations or vaccines had no major impact on the metabolic phenotype of Mo. However, it should be emphasized that in this study population the number of vaccinations and vaccines were virtually non-varying parameters due to vaccination of all individuals at comparable ages according to the national

**Fig. 1 | The differential response of AD and NEO Mo to LPS stimulation is linked to an age-specific baseline programming of metabolism. a** Experimental setup and workflow for blood samples obtained from healthy newborns (NEO) and adults (AD) (each $n = 3$). Created in BioRender. Holsten, L. (2025) https://BioRender.com/c95u965. **b** PCA of the transcriptome data depicting the group relationship of NEO and AD Mo without (Ctrl) and with 4 h treatment with LPS (100 ng/ml). Proportion of variance in percent. **c** Module heatmap resulting from hCoCena colored by the mean GFC within the respective modules and age and treatment groups. **d** GSEA for the modules identified in c, selected Hallmark (HM), gene ontology (GO) and KEGG (KG) terms shown (complete table of enrichment in Supplementary Data 1). **e** TNF and **f** GM-CSF production by untreated (Ctrl) and 16 h LPS-treated (100 ng/ml) NEO Mo (each $n = 8$) and AD Mo (each $n = 8$) represented as means ± SEM. The p-values were determined using one-way ANOVA and *post hoc* Tukey's multiple comparison tests. ns, not significant. **g**–**i** Untreated NEO and AD Mo (each $n = 3$) were cultivated in U-[13 C]-glucose tracer medium and isotopic enrichment determined after 24 h using GC–MS. **g** Scheme of glucose m + 6 metabolism and isotopic enrichment. ETC, electron transfer chain. Created in BioRender. Tödtmann, A. (2025) https://BioRender.com/e96w997. **h** Relative abundance of m + 3 and m + 2 isotopologues of indicated glycolytic pathway and **i** TCA intermediates. Plotted are means ± SEM. The p-values were determined using two-sided t-tests. Source data are provided as a Source Data file.

immunization program, impeding the disclosure of a potential impact. Infections were insufficiently documented and validated in the study population to allow reliable data integration. Only the number of siblings had a decreasing effect on the OXPHOS activity (Fig. 2k), albeit low compared to the influence of age. Still, assuming that a high number of siblings is associated with a high microbial turnover in families[12–16], this finding indicated that microbial exposure in early life might promote metabolic reprogramming towards an AD-like phenotype.

## LPS exposure induces a metabolic shift in NEO Mo towards an AD-like phenotype

Next, the impact of microbial stimulation on the metabolism of human Mo was studied by ex vivo treatment of NEO and AD Mo with LPS and subsequent metabolic profiling. In both NEO and AD Mo, LPS treatment for up to 24 h did not significantly change total energy metabolism, which thus remained higher in NEO compared to AD Mo (Supplementary Fig. 3a). However, LPS treatment had an opposite effect on the dependence of OXPHOS significantly decreasing over time in NEO to AD-like baseline levels, while increasing in AD Mo (Fig. 3a). This opposite effect resulted in a reversed pattern compared to the baseline state with the dependence on OXPHOS being higher in AD than NEO Mo after LPS stimulation (Supplementary Fig. 3b). In contrast, LPS stimulation increased glycolysis in NEO Mo only slightly with strong inter-individual variance but in AD Mo uniformly and highly significant (Fig. 3b). Thus, the glycolytic dependence in NEO converged to AD basal levels following LPS treatment but still remained lower than in LPS-treated AD Mo (Supplementary Fig. 3c). Summarized, these studies suggested that a short-term single LPS treatment induced a metabolic reprogramming of NEO Mo towards an AD-like baseline phenotype, particularly by decreasing OXPHOS and slightly increasing glycolysis. In contrast, AD Mo responded to LPS with a strong activation of both OXPHOS and glycolysis.

To test whether the LPS-induced metabolic shift in NEO Mo also comes with a functional maturation towards an AD-like immune phenotype, we studied the effect of a repeated LPS exposure on marker cytokine responses. In NEO Mo, long-term low-dose LPS pretreatment indeed improved the inflammatory responsiveness to a second challenge with LPS as reflected by a stronger induction of TNF secretion (Fig. 3c), suggesting that continuous low-dose LPS exposure promotes the glycolytic activity of NEO Mo. Opposite, the inducibility of GM-CSF as measure of cell differentiation activity was significantly inhibited in NEO Mo following LPS pretreatment (Fig. 3d), which is in line with LPS decreasing the OXPHOS dependence of NEO Mo. In contrast, LPS-pretreatment of AD Mo impaired the subsequent LPS inducibility of TNF (Fig. 3c) and had no significant impact on the inducibility of GM-CSF (Fig. 3d). Interestingly, the secretion of the regulatory cytokine IL-10, which was like TNF less LPS-inducible in NEO Mo than in AD Mo, was not affected by LPS-pretreatment, neither in AD nor in NEO Mo (Fig. 3e). The findings in AD Mo thus corresponded to classical LPS tolerance and suggested that LPS tolerance is achieved when a primary LPS activation induces a concurrent activation of both OXPHOS and glycolysis (Fig. 3a, b).

Metabolic blocking experiments corroborated our assumption that a strong inflammatory response of AD Mo to a primary LPS stimulus and NEO Mo to a secondary LPS stimulus after LPS priming is linked to their glycolytic activity. In line with the low glycolytic activity of untreated NEO Mo, blocking glycolysis suppressed their TNF response but to a lesser extent and not significantly as in untreated AD Mo (Fig. 3f). In contrast, after LPS pretreatment, the then enhanced TNF response of NEO Mo to secondary LPS activation was significantly inhibited when glycolysis was blocked, while no effect was detectable in LPS-pretreated and thus tolerized AD Mo (Fig. 3g). Blocking OXPHOS had no clear effects on TNF responses neither on the primary or secondary responses of NEO Mo nor that of AD Mo (Fig. 3h, i).

## Ketogenic diet cannot induce a NEO-like immunometabolic phenotype in AD Mo

Ketogenesis appears to be an integral part of extrauterine metabolic adaptation in the term human neonate, as from 12 h of age, healthy term infants show high ketone body turnover rates approaching those found in adults after several days of fasting[17–19]. All infants included in our study were exclusively breastfed for the first six months of life. Breast milk is a high-fat and slightly carbohydrate-restricted diet (Supplementary Table 3). It has been speculated that augmented ketogenesis in breast-fed infants may be due to the activation of mitochondrial fatty acid b-oxidation by breast milk factors[20].

To investigate whether ketosis induces a NEO-like immunometabolic phenotype in AD Mo, 10 healthy adult volunteers underwent an eucaloric KD for one week (Supplementary Table 2). The immunometabolic profile of blood Mo was determined under carbohydrate-rich western diet conditions before (Pre KD) and on day 7 after starting KD (Post KD) (Fig. 4a). One week of KD was chosen since KD in adults induced significant ketosis from day 2 on, so that after 7 days of KD ketosis prevailed for at least 5 days (Fig. 4b). This time period corresponded with the time period of elevated ketone body concentrations in neonates during the first 5–7 days of life[21,22]. KD did not induce a NEO-like increase but oppositely a significant decrease of baseline metabolic activity in AD Mo (Fig. 4c). This was associated with a significant reduction of the dependence on glycolysis (Fig. 4d) but did not change the dependence on OXPHOS (Fig. 4e), and did therefore not achieve an adoption of a NEO-like metabolic state. Also different from NEO Mo (Fig. 3, Supplementary Fig. 3), LPS stimulation of adult Post KD Mo led to a consistent increase of the energy metabolism and glycolysis (Supplementary Fig. 4a, b), in fact to levels comparable with that of Pre KD Mo (Fig. 4c, d), whereas OXPHOS remained unaffected (not decreasing like in NEO Mo) (Fig. 4e, Supplementary Fig. 4c). Accordingly, KD neither changed the inducibility of TNF (Fig. 4e) or GM-CSF (Fig. 4f) in LPS-activated AD Mo.

Thus, there is no evidence for different diets in neonates and adults explaining the immunometabolic differences between their Mo, which further supports the concept that the immunometabolism of Mo primarily follows a developmentally-driven unidirectional reprogramming linked to age and environmental cues such as microbial exposures.

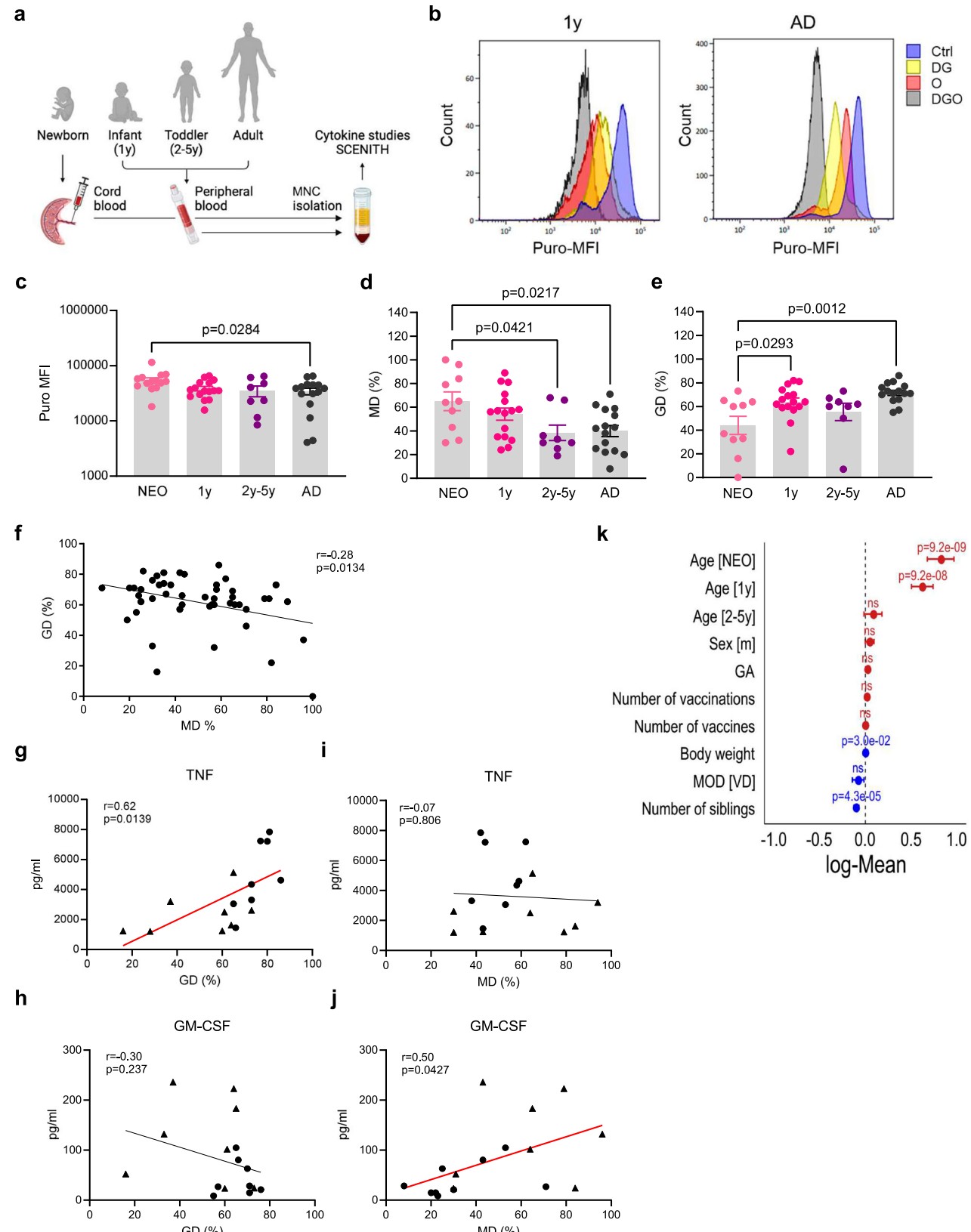

## Control of baseline myeloid differentiation depends on OXPHOS

The transcriptomic and immunometabolic data suggested that high OXPHOS activity in NEO Mo is associated with ongoing cell differentiation processes, whereas AD Mo primarily spend energy on inflammatory responsiveness with glycolysis as main energy source.

To elucidate whether myeloid differentiation under baseline conditions differs between NEO and AD Mo, we let them differentiate ex vivo into macrophages (MDM) in the presence of human AD blood plasma, without additional cytokine-driven polarization. Unexpectedly, significantly more AD than NEO Mo survived during

**Fig. 2 | Metabolic reprogramming during the first years of life shifts Mo from high OXPHOS to high glycolytic dependence. a** Experimental setup and workflow for blood samples obtained from healthy newborns (NEO), infants (1 y), toddlers and preschoolers (2–5 y), and adults (AD). Created in BioRender. Ehlers, G. (2025) https://BioRender.com/f67m295. **b** Representative flowcytometric SCENITH studies in infant and AD untreated (Ctrl) or metabolically blocked (DG (2-Deoxy-D-Glucose) or O (oligomycin) or both (DGO)) MNCs showing the mean fluorescence intensity of puromycin (Puro-MFI) of the Mo population. **c** Age dependent energy metabolism (Puro MFI), and percentages of **d** OXPHOS dependence (MD, mitochondrial dependence) and (**e**) glycolytic dependence (GD) of Mo (NEO: $n = 10$; 1 y: $n = 16$; 2–5 y: $n = 8$; AD: $n = 15$) at baseline plotted as means ± SEM. Significant differences were determined using one-way ANOVA and *post hoc* Tukey's multiple comparison tests. **f** Correlation between baseline OXPHOS and glycolytic dependence in Mo obtained from the entire study population. Indicated are best fit regression lines, $r$, Pearson's correlation coefficients and p-values of correlations. **g–j** Correlations between the LPS-induced production of TNF (**g, i**) and GM-CSF (**h, j**) by NEO (triangles) and AD (dots) Mo and their baseline glycolytic (**g, h**) and OXPHOS (**i,j**) dependence. Indicated are best fit regression lines, $r$, Pearson's correlation coefficients and p-values of correlations. **k** Effect sizes building a generalized linear model of indicated factors potentially influencing the basal OXPHOS dependence in Mo (NEO: $n = 10$; 1 y: $n = 16$; 2–5 y: $n = 8$; AD: $n = 15$) were plotted as log means ± CI. The *p*-values were determined using the Wald test. GA, gestational age, MOD, mode of delivery, VD, vaginal delivery. Source data are provided as a Source Data file.

differentiation (Fig. 5a), yielding higher numbers of AD MDM than NEO MDM (Fig. 5b). This was linked to increased apoptosis and necrosis rates in NEO MDM (Fig. 5c), while neither NEO nor AD MDM proliferated actively (Fig. 5d). However, NEO MDM appeared morphologically further differentiated than AD MDM, showing more irregular cell shapes and prominent cell extensions (Fig. 5e) and larger average cell sizes (Fig. 5e, f). In line with the higher cell death rates, a significant proportion of cell-free nuclei was observed among NEO but not AD MDM (Fig. 5e, g). Furthermore, NEO MDM differentiation resulted in higher proportions of CD68 macrophage marker positive MDMs (Fig. 5h, Supplementary Fig. 5a, b) with increased expression of CD80 (Fig. 5i, Supplementary Fig. 5d) compared to AD MDM, while the ratio of M1-like CCR7[+] and M2-like CCR7[-] subsets[23] was similar (Fig. 5j, Supplementary Fig. 5a–c). These fully differentiated NEO MDM produced also higher levels of TNF and IL-10 (but not IL-6) in response to LPS than AD MDM (Fig. 5k), collectively suggesting higher stemness of NEO Mo compared to AD Mo.

Of note, using high-dose GM-CSF/M-CSF (50 ng/ml) instead of AD blood plasma (with physiological concentrations of GM-CSF ( < 1 to 10 pg/ml) and M-CSF (20–1000 pg/ml)[24], maximized and aligned the yield of well-differentiated MDMs derived from NEO and AD Mo, concealing the age signature of MDM differentiation (Supplementary Fig. 6a–c). Using cord blood plasma instead of AD blood plasma did not change total cell yield, apoptosis rates, and proportions of CD68[+] MDMs significantly but the differences between NEO and AD MDM differentiation became blurry and lost significance, likely due to the higher compositional variability of cord blood[25–27] (Supplementary Fig. 6d–f).

To investigate how the metabolic pathways are linked to myeloid differentiation, we inhibited glycolysis and OXPHOS in NEO and AD Mo seeded for MDM differentiation. Overall, the yield of AD MDM was again higher than in NEO MDM (Fig. 5l). In both NEO and AD MDM, cultivation under low, in fact physiological glucose conditions led to a slight but in the presence of DG to a significant cell loss compared to standard high glucose culture conditions. Oppositely, inhibition of OXPHOS significantly improved the survival of NEO and AD MDM compared to physiological low glucose conditions, even compared to high glucose conditions slightly higher MDM numbers were observed (Fig. 5l), suggesting that OXPHOS promotes while glycolysis inhibits apoptosis and cell death in differentiating Mo. The analysis of apoptosis rates and the yield of proportions of well-differentiated CD68[+] MDMs confirmed an expected increase of both when glycolysis was inhibited, which was particularly consistent in the highly glycolysis-dependent AD cells (Fig. 5m, o). In contrast, apoptosis rates and proportions of CD68[+] MDMs were decreased when OXPHOS was inhibited, however, consistently only in the highly OXPHOS-dependent NEO cells (Fig. 5n, p).

Collectively, these findings show that myeloid differentiation is well controlled in NEO Mo and accompanied by apoptosis-mediated cell sorting that is supported by high OXPHOS activity, resulting in lower macrophage yield but improved phenotypic and functional differentiation. High glycolytic activity like in AD Mo prevents cell death and apoptosis during differentiation into macrophage but at the cost of the macrophage quality.

## Network studies identify potential transcriptional regulators of the ontogeny of Mo immunometabolism

Considering the complexity that two metabolic pathways are reprogrammed with age in an inverse and mutual manner raises the question which transcription factors (TFs) in human Mo regulate the balance between glycolytic and OXPHOS activity and related functions, i.e. control of myeloid differentiation and inflammatory responsiveness. To address this question, we first determined which gene modules differed most significantly in mean baseline gene expression between NEO and AD Mo and identified the modules maroon, steelblue, and lightgreen (Fig. 6a, Supplementary Fig. 7a). Following our concept that the cell metabolism determines Mo functions, we defined those transcription factors (TFs) as top candidate regulators, whose binding sites (TFBS) were enriched in the proximity of metabolism-related steelblue genes and additionally and either maroon genes (cell cycle control and differentiation) or lightgreen genes (inflammatory response) or both. This approach identified a total of 16 candidate TFs (Fig. 6b, Supplementary Table 4), of which *E2F1*, *MYB*, *STAT1*, and *FLI1* additionally proved to be among the top fifteen hub genes within their respective network module (Fig. 6c). Besides fulfilling the basic prerequisite of being regulators of metabolism-related (steelblue) genes, all of them were additionally predicted regulators of inflammatory response-related (lightgreen) genes. *E2F1* was the only TF whose binding sites were also enriched in the proximity of cell differentiation-related (maroon) genes (Fig. 6b, Supplementary Table 4). In fact, about two-third of *E2F1* target genes belonged to lightgreen genes and one-third to maroon and steelblue genes, respectively (Supplementary Fig. 7b). Network construction based on all 16 candidate TFs and their target TFs also positioned *E2F1*, *MYB*, *STAT1* and *FLI1* as central TFs connecting all three functional modules by regulating transcriptional regulators of the other modules (Fig. 6d, Supplementary Fig. 7c). Importantly, *E2F1*, *MYB*, and *STAT1* were also the only TFs differentially expressed in NEO and AD Mo, namely *E2F1* and *MYB* higher in NEO than AD Mo and oppositely *STAT1* higher in AD than NEO Mo (Fig. 6b, Supplementary Tables 3 and 5).

Collectively, based on their module affiliations, central positions in the entire regulatory network, and differential expression, the data point to central roles for *E2F1* and *MYB* as drivers of the NEO immunometabolic phenotype, whereas the adult immunometabolic phenotype appears to be guided by *STAT1*, and potentially also by *FLI1*.

## *E2F1*, *MYB*, *STAT1* and *FLI1* are important regulators of the age-dependent immunometabolism of Mo

To validate age-dependent expression and gain further evidence for a connection to the metabolic programming, we studied the age dependence of *E2F1*, *MYB*, *STAT1* and *FLI1* expression longitudinally using Mo isolated from blood sample of healthy newborns, infants, toddlers, and adults. Both *E2F1* and *MYB* were expressed at high levels in newborn Mo that steadily decreased to AD-like levels by age 2–3 years (Fig. 7a). Their expression profiles thus strongly resembled the

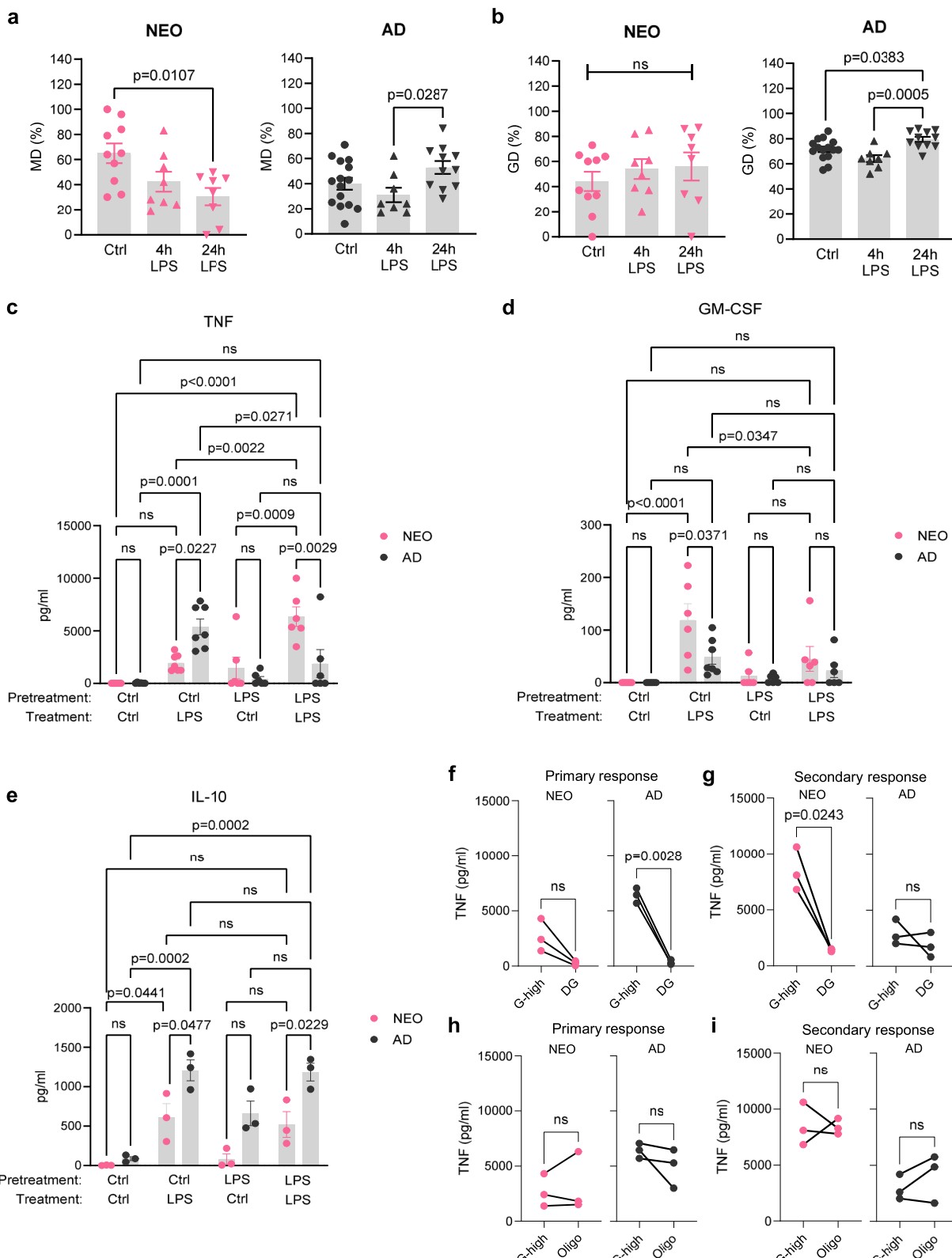

gradual decrease of OXPHOS dependence, which also reached AD-like levels only at toddler age (Fig. 2d). In contrast, the expression of *STAT1* was low in newborn Mo and then increased within the first year of life to AD-like levels (Fig. 7a). This was strikingly similar to the age-dependent progression of glycolysis dependence, which also became AD-like within one year after birth (Fig. 2e). *FLI1* expression did not

significantly change over age (Fig. 7a), yet it still might support regulatory functions of *STAT1* as *STAT1* is its direct target of regulation (Fig. 6d, Supplementary Fig. 7c).

To test the effect of overexpression of *E2F1*, *MYB*, *STAT1* and *FLI1* on predicted functions at a population-based level, we performed a human variation analysis (HUVA)[28]. This recently introduced

**Fig. 3 | LPS exposure induces a metabolic shift in NEO Mo towards an AD-like phenotype. a** OXPHOS and (**b**) glycolytic dependence in NEO and AD Mo over time of LPS activation assessed by SCENITH studies in NEO and AD MNCs treated for 4 h (each *n* = 8) and 24 h (NEO *n* = 8, AD *n* = 15) with LPS (1 μg/ml) compared to baseline (untreated Ctrl; NEO *n* = 10, AD *n* = 11). Plotted are means ± SEM. Significant differences were determined using one-way ANOVA and *post hoc* Tukey's multiple comparison tests. **c–e** Isolated NEO and AD Mo were control or LPS (0.1 ng/ml) pretreated for 16 h followed by activation with 1 μg/ml LPS for 4 h. **c** TNF (NEO *n* = 6,

AD *n* = 7), **d** GM-CSF (NEO *n* = 6, AD *n* = 7), and (**e**) IL-10 (each *n* = 3) production represented as means ± SEM. *p*-values were determined using one-way ANOVA and *post hoc* Tukey's multiple comparison tests. **f–i** TNF production by not pretreated (**f, h**; primary response) and 16 h LPS (0.1 ng/ml) pretreated (**g, i**; secondary response) NEO and AD Mo (each *n* = 3) upon activation with 1 μg/ml LPS for 4 h without (G-high) and in the presence of DG (**f, g**) or oligomycin (Oligo) (**h, i**). p-values were determined using paired two-sided t-tests. Source data are provided as a Source Data file.

computational approach allows to contrast adult individuals with highest *versus* lowest gene of interest expression within the Human Functional Genomics Project (FG500 cohort, 96 whole blood RNAseq samples of healthy adult volunteers)[29] (Fig. 7b). Individuals within the top and bottom 0.1 quantiles of gene of interest expression in circulating immune cells (Fig. 7c, f, i, l) proved to show highly differential *E2F1*, *MYB*, *STAT1* and *FLI1* expression (Fig. 7d, g, j, m). Interestingly, and similar to the situation in NEO *versus* AD Mo, *E2F1* and *FLI1* were oppositely (Fig. 7d, m) and *STAT1* and *FLI1* concordantly (Fig. 7j, m) differentially expressed in their high *versus* low donor subgroups, pointing again to contrary (*E2F1 versus FLI1*) respective similar (*STAT1* and *FLI1*) regulatory functions. High and low expressing donors were then contrasted and co-regulated genes interrogated for functions (GSEA based on FC ranked gene lists). This revealed an enrichment of genes involved in OXPHOS, cell cycle control, and myeloid cell differentiation in *E2F1* and *MYB* high individuals, and even a negative enrichment of inflammatory response genes in *MYB* high individuals (Fig. 7e, h). This further established *E2F1* and *MYB* as important regulators of the NEO Mo phenotype and suggested that *E2F1* and *MYB* high expressing adults are characterized by a NEO-like immunometabolism. In contrast, co-regulated genes in *STAT1* and *FLI1* high individuals were positively enriched for inflammatory response genes but negative for OXPHOS-related genes, and *FLI1* high individuals even negative for cell differentiation-related genes (Fig. 7k, n), supporting *STAT1* and *FLI1* being drivers of the AD Mo phenotype.

To assess the effect of inhibition of the candidate TFs on predicted functions, we avoided genetic editing of Mo to not distort their primary transcriptional and metabolic programming but decided to employ pharmacological inhibitors. Therefore, AD Mo were treated with either HLM006474 (HLM) inhibiting *E2F1*[30,31], all-trans retinoic acid (ATRA) inhibiting *MYB*[32,33], fludarabine (FA) inhibiting *STAT1*[34,35], and camptothecin (CPT) inhibiting *FLI1*[36,37]. Even though these inhibitors might have broader effects than exclusively inhibiting *E2F1*, *MYB*, *STAT1*, and *FLI1*, respectively, the consequences for metabolic pathways activities and MDM differentiation largely supported the computationally predicted TF functions. Inhibition of *E2F1* slightly decreased the OXPHOS activity of Mo (Fig. 8a) while glycolysis remained largely unaffected (Fig. 8b). With respect to the outcome of MDM differentiation, *E2F1* inhibition yielded higher numbers of MDMs (Fig. 8c) but lower proportions of well-differentiated CD68 macrophage marker positive MDMs (Fig. 8e), while no effect on apoptosis was observed (Fig. 8d). *MYB* inhibition led to a significant downregulation of OXPHOS but also activated glycolysis (Fig. 8a, b), which together hampered MDM differentiation profoundly as reflected by low MDM yield, increased apoptosis rates and strongly reduced numbers of CD68-expressing MDMs (Fig. 8c–e). Inhibition of *STAT1* had no clear effect on Mo cell metabolism while *FLI1* inhibition led to a significant decrease in glycolysis but no perceivable effect on OXPHOS (Fig. 8a, b). However, inhibition of both *STAT1* and *FLI1* hampered MDM yield due strongly enhanced apoptosis but did not impair the quality of MDM differentiation in terms of proportions of CD68+ cells (Fig. 8c–e). Thus, in essence, *E2F1* and *MYB* concordantly promoted OXPHOS and high-quality myeloid differentiation. In contrast, inhibition of *MYB*, *STAT1* and *FLI1* promoted cell death during myeloid differentiation, which was therefore unlikely linked to glycolysis since glycolysis was inhibited by *MYB* but promoted by *FLI1*.

Collectively, our findings strongly support a model of *E2F1*, *MYB*, *STAT1* and *FLI1* being important regulators of the postnatal ontogeny of Mo immunometabolism with high *E2F1* and *MYB* expression in NEO Mo promoting OXPHOS and cell differentiation that shifts to low *E2F1* and *MYB* expression and high *STAT1* expression with increasing age, driving glycolysis and inflammatory activity in AD Mo in liaison with *FLI1*.

## Discussion

The restriction of innate immune responses in neonates is a conundrum. It has been associated with an increased risk of invasive infections but also suggested as neonate-specific physiological programming preventing overshooting responses to the new extrauterine environment[5,38]. In adults, the importance of a fast adaptability of energy metabolism to meet the demands of immune cells responding to microbial challenges has repeatedly been demonstrated[6,7]. However, the physiology of the immunometabolism in neonates remains elusive. By combining comprehensive systems biology approaches and functional studies, we revealed hallmarks of the postnatal ontogeny of immunometabolism in human Mo that might change our understanding of how age and environmental cues influence metabolic programming and related cell functions.

In line with published correlative data in AD and NEO Mo and macrophages[8,9], we demonstrated by direct linkage of metabolic and cellular functions that the reduced inflammatory responsiveness of NEO Mo results from a reduced basal glycolytic activity. This recapitulates the immunometabolic coupling in AD Mo, where rapid energy demand for immediate inflammatory response to microbial challenges is met fastest by glycolysis though producing less ATP than OXPHOS[6,7]. However, we showed that reduced glycolysis in NEO Mo comes with an increased OXPHOS activity that provides high energy supply for ongoing myeloid differentiation processes in NEO Mo involving apoptosis, cell cycle control, and chromatin remodeling. Recently, macrophages in proliferation states (S-G2/M-phase) have been shown to be less plastic and sensitive to interferon-g-induced polarization[39]. This is well in line with our finding of high expression of genes involved in cell cycle control by NEO Mo, particularly G2/M checkpoint regulators, that might warrant high plasticity and high-quality myeloid differentiation. A multitude of findings well explain the generally higher cell differentiation activity of NEO Mo, e.g., ongoing fetal growth hormonal cues[40], differentiation and expansion of nonclassical Mo after birth[41,42], or colonization of tissues with NEO Mo that differentiate into tissue-resident macrophages[43–45]. Murine in vitro and in vivo studies support the linkage between OXPHOS and myeloid differentiation showing that an inhibition of fatty acid β-oxidation, which primarily fuels OXPHOS, impairs Mo differentiation into dendritic cells and macrophages[46]. Furthermore, insulin-like growth factor 2, one of the most important drivers and markers of intrauterine and postnatal growth and differentiation[40], was reported to induce persistent OXPHOS in maturing murine macrophages keeping them in an anti-inflammatory state[47]. Therefore, considering the general developmental state of neonates and these major tasks of NEO Mo, it is plausible that high OXPHOS activity in NEO Mo is a physiological program at the cost of glycolytic and related inflammatory activity. Accordingly, our findings suggest that shifting the balance at this age prematurely towards glycolysis could increase the risk of hyperinflammation and inhibit essential OXPHOS and related myeloid

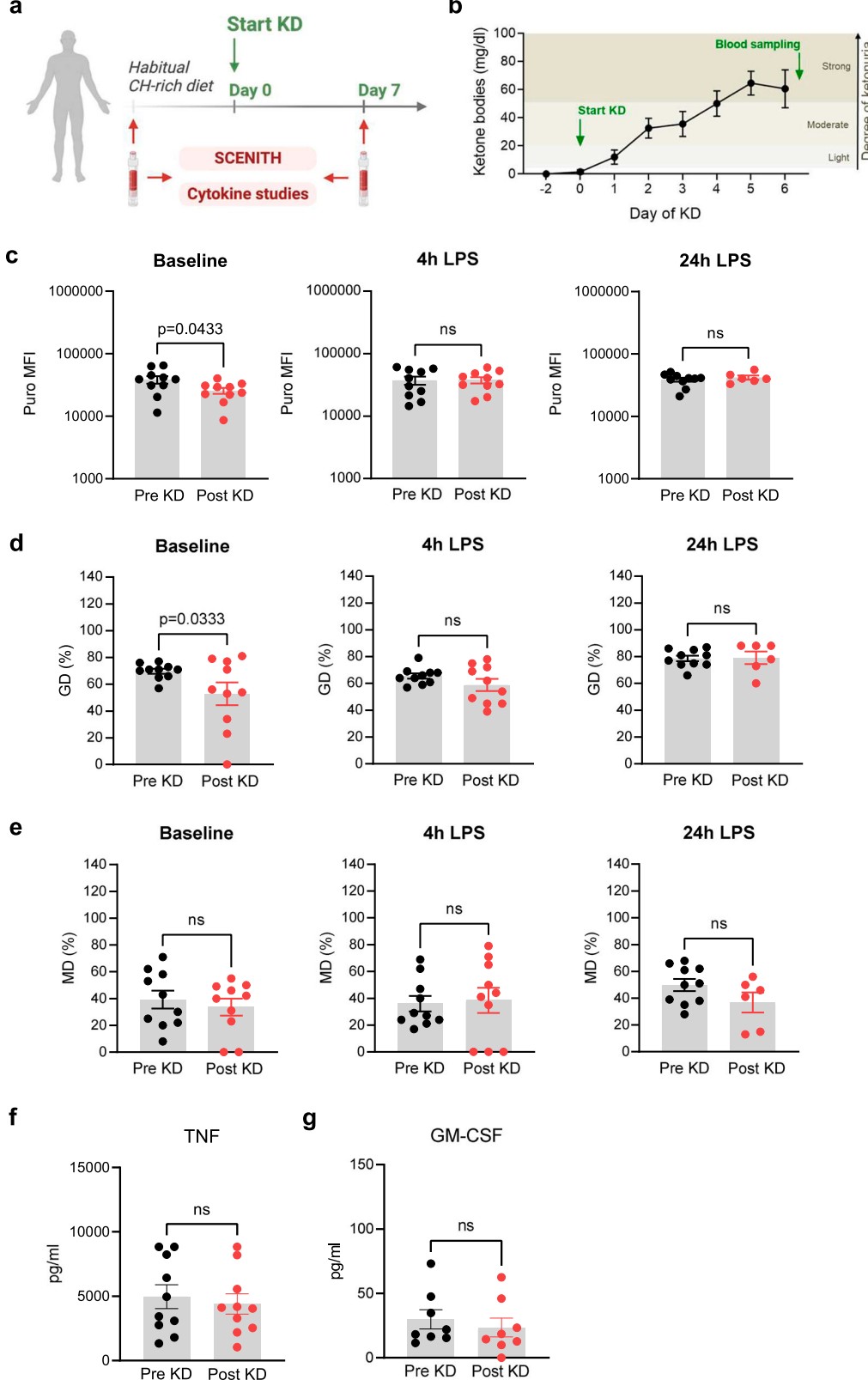

**Fig. 4 | Ketogenic diet cannot induce a NEO-like immunometabolic phenotype in AD Mo. a** Experimental setup and workflow of blood samples obtained from 10 healthy adults before and 7 days after start of ketogenic diet (KD). Created in BioRender. Ehlers, G. (2025) https://BioRender.com/f67m295. **b** Urine ketone levels before and during the course of KD in the 10 healthy adults. Direct comparison of (**c**) energy metabolism, (**d**) glycolytic and (**e**) OXPHOS dependence of Pre-KD and Post-KD Mo ($n = 10$ per group) assessed by SCENITH studies in MNCs at baseline and after 4 h and 24 h of LPS stimulation (1 µg/ml). **f** TNF and (**g**) GM-CSF production by isolated Pre-KD and Post-KD Mo ($n = 10$ per group) upon LPS stimulation (100 ng/ml) for 16 h. Plotted are means ± SEM. *p*-values were determined using two-sided MWU-tests. Source data are provided as a Source Data file.

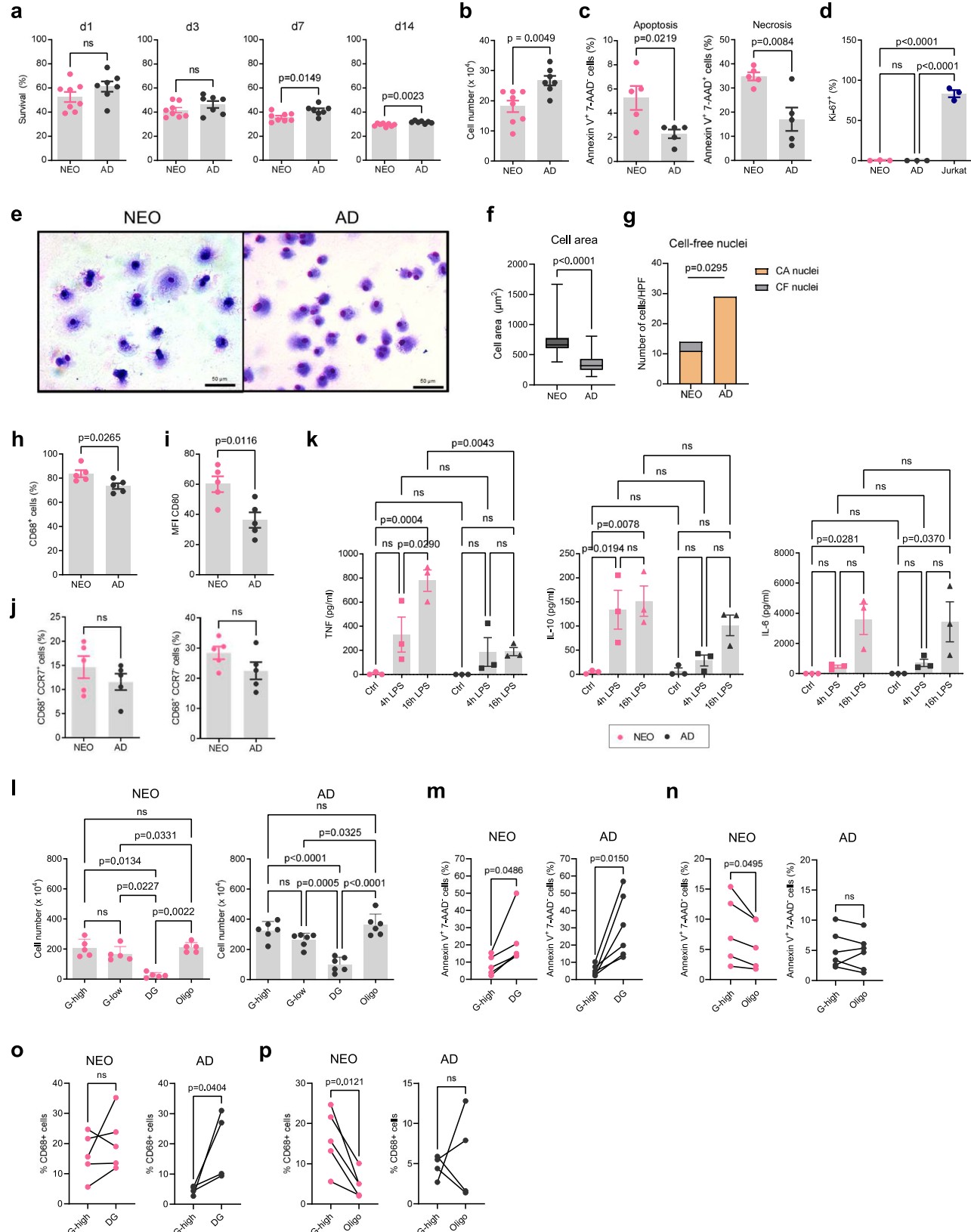

differentiation, which both would promote fatal courses of sepsis. This is supported by recent clinical observations in preterm infants and in a preterm piglet model both showing that increased parental glucose supply significantly increases the risk of fatal sepsis[48,49]. Acknowledging that the immune responsiveness of Mo depends inversely on the activity of cell differentiation processes that are in turn primarily a

function of age, it is conclusive that a KD in adults could not significantly increase OXPHOS in AD Mo and therefore not induce a NEO-like metabolic phenotype.

In this work, we were interested in the age-specific cell-intrinsic myeloid differentiation capacity and therefore used standardized culture conditions with pooled AD blood plasma. This approach

**Fig. 5 | Control of baseline myeloid differentiation depends on OXPHOS.**
**a** Percent of surviving NEO ($n = 8$) and AD ($n = 7$) Mo at indicated time points during ex vivo differentiation into macrophages (MDM) in the presence of human AD blood plasma. **b** MDM numbers after 14 days of differentiation of equal numbers of NEO ($n = 8$) and AD ($n = 7$) Mo. **c** Proportion of apoptotic (Annexin V$^+$7-AAD$^-$) and necrotic (Annexin V$^+$7-AAD$^+$) cells among NEO and AD MDM (each $n = 5$). Plotted are means ± SEM. p-values were determined using two-sided MWU-tests. **d** Proportion of Ki67$^+$ proliferating cells among NEO and AD Mo treated for 24 h with GM-CSF and Jurkat cells as positive control represented as means ± SEM (each $n = 3$). *p*-values were determined using one-way ANOVA and *post hoc* Tukey's multiple comparison test. **e** Representative images of the morphology of NEO and AD MDM. May-Grunwald-Giemsa-staining. Scale bars, 50 μm. **f** Mean of cell areas counted in 4 HPFs of $n = 2$ samples, respectively. Box plots show medians (central line), interquartile ranges (box edges) and whiskers extending to the smallest and largest data points. The p-value was determined using a two-sided MWU-test. **g** Mean number of cell-associated (CA) and cell-free (CF) nuclei per HPF. The p-value was determined using

a two-sided Fisher's exact test. **h** Proportion of CD68$^+$ MDM after 14 days of differentiation. **i** CD80 expression (MFI) in CD68$^+$ MDM. **j** Proportion of CCR7$^+$ and CCR7$^-$ cells from CD68$^+$ MDM. Plotted are means ± SEM (each $n = 5$). p-values were determined using two-sided MWU-tests. **k** Production of indicated cytokines by control- and 4 h and 16 h LPS-treated (10 ng/ml) NEO and AD MDM (each $n = 3$). Plotted are means ± SEM. Significant differences were determined using one-way ANOVA and *post hoc* Tukey's multiple comparison tests. **l–n** NEO and AD Mo were cultivated for 24 h in the presence of 300 mg/dL (G-high) or 100 mg/dL glucose (G-low) or treated with DG or Oligo under G-high conditions. **l** Cell numbers after cultivation (NEO $n = 5$, AD $n = 6$). Plotted are means ± SEM. *p*-values were determined using one-way ANOVA and *post hoc* Tukey's multiple comparison tests. **m–p** Proportions of apoptotic cells (NEO $n = 5$, AD $n = 6$) (**m, n**) and CD68$^+$ MDM (NEO $n = 5$, AD $n = 4$) (d3) (**o, p**) after cultivation in the presence of DG (**m, o**) or Oligo (**n, p**) compared to standard G-high conditions, respectively. p-values were determined using paired two-sided t-tests. Source data are provided as a Source Data file.

neglects potential age-specific influences from the tissue microenvironment on macrophage differentiation in vivo. Using blood plasma is presumably generally not appropriate to model tissue microenvironments, particularly not that of neonates when using cord blood plasma given its dramatic compositional changes within the first days after birth[50]. Nevertheless, we tested the effect of cord blood plasma and observed that MDM differentiation tended to be less supported than by AD blood plasma and lost significance regarding differences between NEO and AD MDM. We ascribe this to the high compositional variability of cord blood plasma[25–27,50] that can be rich in factors promoting (e.g. GM-CSF) as well as regulating myeloid differentiation (e.g. TGF-β)[51,52].

Our study revealed that the postnatal increase of glycolysis in Mo takes place within the first year of life with microbial exposures being likely the most important drivers for the activation of glycolysis. The major microbial exposures accumulating in the first year of life are the colonizing microbiota, vaccines, and natural infections, which are all well-known to additively and/or synergistically promote innate immune functions in children. The direct association between early-life exposures to microbial triggers and the cellular metabolism has not been studied yet in vivo. We found a small though significant decreasing sibling effect on the OXPHOS activity, preliminarily suggesting a link to a more diverse microbial exposure[12–16]. A recent study in newborn infants found a strong association between BCG vaccination-induced decreases of plasma lysophospholipid levels, particularly of lysophosphatidylcholine metabolites (LPCs), and enhanced blood cytokine inducibility in vitro[53]. Of note, since LPCs are metabolites of phosphatidylcholine, the major component of cellular membranes and marker of high cell turnover[54], this might point to a BCG-induced increased immune reactivity at the cost of cell growth and differentiation processes.

Microbial pretreatment of innate immune cells can have two different functional consequences for subsequent antimicrobial responses, either enhancement based on "priming" or "trained immunity" or containment resulting in "immune tolerance" or "immunoparalysis"[55]. In our study, LPS pretreatment had age-dependently opposite effects and enhanced subsequent LPS responses in NEO Mo but paralyzed those of AD Mo, suggesting that the initial state of metabolic programming might determine the outcome of LPS pretreatment. The higher baseline glycolytic activity in AD Mo is likely the result of previous microbial exposures[56,57] and possibly not further gradable in contrast to the low baseline glycolytic activity in largely naïve NEO Mo. However, the even more important key to the opposite effects of LPS pretreatment in AD and NEO Mo might be the differential OXPHOS activity. A high OXPHOS activity like in NEO Mo prevents the accumulation of TCA metabolites such as α-ketoglutarate and itaconate[58] and thus induction of immune tolerance[59–61]. In addition, high OXPHOS activity leads to an accumulation of malate and fumarate, both promoting innate immune training[62]. The concept of high OXPHOS

activity being the basis for a strong priming effect on innate immunity is further corroborated by our observation that the OXPHOS activity is still higher in Mo of one-year aged infants than adults, as the innate immune system is still more plastic and receptive for imprinting cues at infancy than at adulthood[63].

Based on TF network construction and overexpression studies at a population-based level, *E2F1*, *MYB*, *STAT1*, and *FLI1* were predicted as central regulators balancing Mo metabolism to meet age-specific functions. Their expression profiles over age together with TF inhibition experiments suggested that *E2F1* and *MYB* drive the NEO Mo phenotype by promoting cell cycle control and myeloid differentiation as well as OXPHOS. Both has been shown for *E2F1*, its critical function for myeloid development[64–66] and its promoting influence on OXPHOS by regulating the supercomplex assembly factor HIG2A[67]. Moreover, there is compelling evidence that *E2F1* induces apoptosis and triggers the elimination of potentially hyperplastic cells[64,68–70], supporting our finding that NEO Mo control myeloid differentiation by OXPHOS-dependent apoptotic cell sorting. *MYB* is also a known key regulator of hematopoietic cell proliferation and differentiation[71], in particular the differentiation of fetal Mo[72], and has been implicated in mediating a metabolic switch from glycolysis to OXPHOS[73]. Opposite, our study pointed to *STAT1* and *FLI1* being the drivers of the AD immunometabolic Mo phenotype by promoting inflammatory responsiveness at the cost of myeloid differentiation control. This is in line with the strongly proinflammatory skewed Mo phenotype in patients with *STAT1* gain-of-function mutations[74]. Moreover, though we could not detect an effect of *STAT1* inhibition on the cell metabolism in Mo, *STAT1* opposite to *E2F1* and *MYB* has been shown to promote glycolysis during M1 macrophage polarization[75]. The direct impact of *FLI1* on glycolysis and OXPHOS has not been investigated so far but our studies revealed a promoting effect on glycolysis, in line with its known effect to enhance inflammatory LPS responses of human Mo and macrophages[76]. The strong inhibitory effect found for *FLI1* and *STAT1* on apoptosis during MDM differentiation is in line with the apoptosis repressive effect of *STAT1* in activated macrophages shown by others[77]. Collectively, this supports the concept that myeloid differentiation in neonates is controlled by OXPHOS-dependent apoptotic cell sorting as long as there is no need for enhanced inflammatory responsiveness.

In summary, our data suggest that the ontogeny of the immunometabolism of human Mo progresses in a unidirectional manner that is tightly regulated to ensure an age-appropriate balancing of energy demands and cell functions. We propose a model according to which the cell metabolism of Mo switches during early childhood from OXPHOS towards glycolysis to support age-specific Mo tasks. While myeloid differentiation requiring OXPHOS-derived high energy supply is a central mission of newborn Mo, childhood comes along with increasing exposures to microbial challenges that induce an essential switch to glycolysis to ensure fast and strong immune responsiveness.

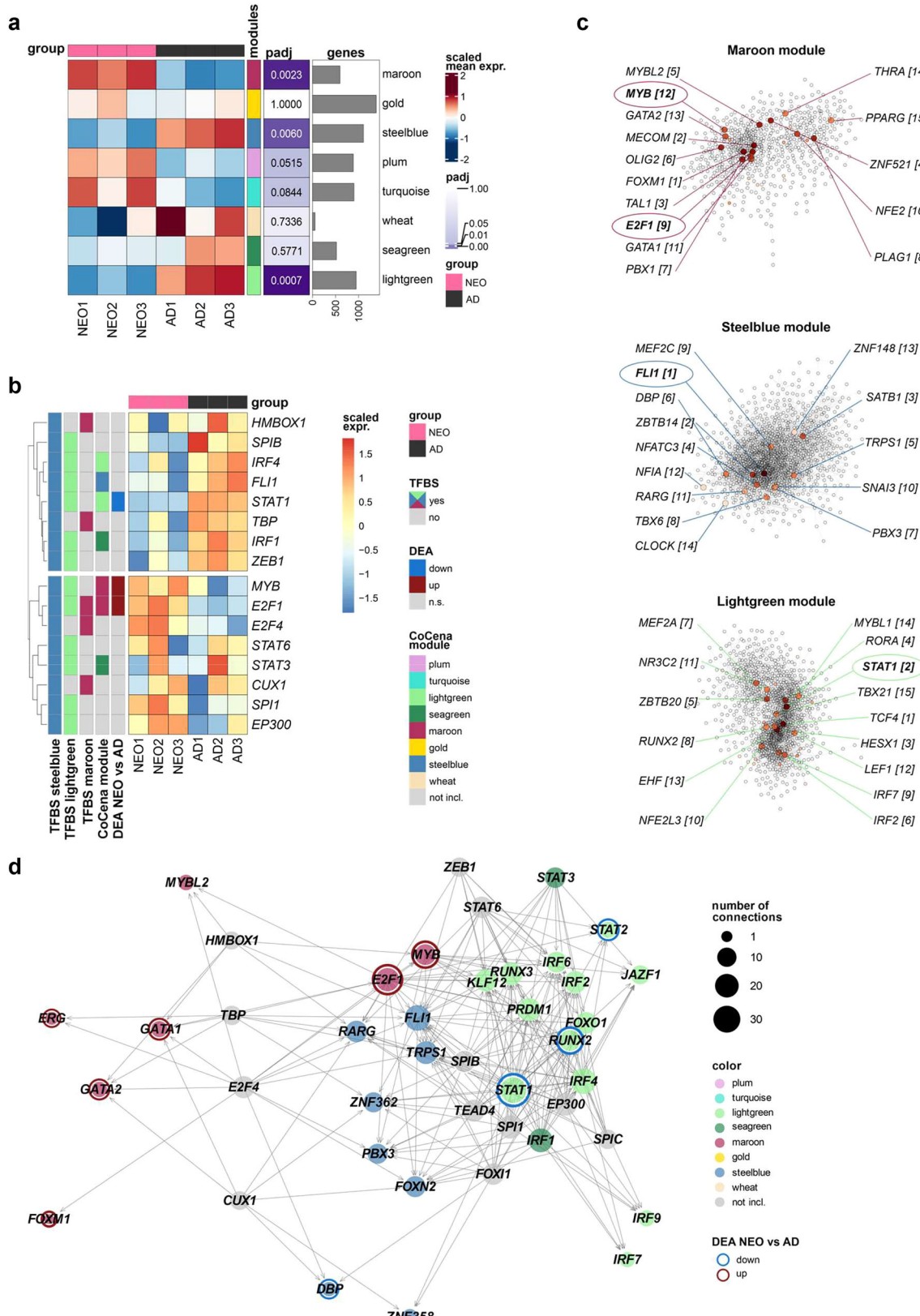

## Methods

### Study population

Study participants were enrolled between January 2021 and September 2024 at the Hannover Medical School (Hannover, Germany) and between January 2022 and September 2024 at the University Hospital Würzburg (Würzburg, Germany) in the frame of the MIAI birth cohort study[78] after

written informed consent was obtained from parents and AD donors, respectively. For the transcriptomic analysis and isotope tracing studies, cord blood of term newborn infants and peripheral blood of AD volunteer donors aged 24–53 years was used. For the functional metabolic studies (SCENITH), cord blood of 10 newborn infants (NEO group) and peripheral blood from 16 breastfed infants (1 y group), 8 toddlers and

**Fig. 6 | Transcriptional regulator network of immunometabolism in Mo.**
**a** Module heatmap of the scaled mean expression of genes within the respective network modules for each donor. Bonferroni adjusted *P* (padj) of the two-sided t-test comparing the mean expression per module in the group NEO *versus* AD. **b** Heatmap of selected TFs identified by TFBS overrepresentation analysis as potential regulators of the steelblue module and additionally the maroon and/or lightgreen module. Affiliations to hCoCena modules and significant expression differences are color-coded. **c** Network visualization with hub genes of the maroon,

steelblue and lightgreen module highlighting the top 15 hub TFs, ranks in brackets. Color-highlighted TFs belong to the group of candidate TFs depicted in (**b**). **d** Network construction of selected TFs shown in (**b**) and their target TFs colored by hCoCena module affiliation or in grey if not included within the network. Node sizes are scaled by the number of connections. Red and blue ring colors, respectively, highlight the TFs significantly higher or lower expressed in NEO *versus* AD Mo. Source data are provided as a Source Data file.

preschoolers (2–5 y group), and 15 AD volunteer donors (AD group) were collected (Table S1). For gene expression studies (qRT-PCRs), peripheral blood from 11 newborns aged 3 days, 11 infants aged 8–13 months, 6 healthy toddlers aged 2–3 years, and 11 AD volunteer donors aged 28–56 years were collected. Only healthy individuals without a history of inflammation and infection during the last 4 weeks or organ dysfunction were included, respectively. Children from pregnancies that involved births resulting from maternal trauma and children with major anomalies, inborn errors and acute or chronic diseases were excluded from this study. For the ketogenic diet (KD) studies, ketosis was induced in 10 healthy AD volunteers following an eucaloric high-fat KD (carbohydrates < 30 g per day) ad libitum for 1 week (Table S2). Blood sampling and measurements of the participant's body weights and body mass indices (BMI) were done two days before and at day 7 after starting KD. To verify induction of ketosis, urine levels of ketone bodies and glucose were analyzed daily from two days before start until end of KD using Combur test strips (Roche).

### Ethics statement
Protocols and usage of human biomaterial were approved by the Institutional Review Boards of the Hannover Medical School (no. 8014_BO_S_2018, no. 7224-2016, Research Obstetrics Biobank no. 1303-2012) and the University of Würzburg (no. 13/22_z-am). Written informed consent was obtained from all participating individuals and the parents or legal representatives on behalf of the children enrolled in our study.

### Cell isolation and culture conditions
**Human monocytes (Mo) and mononuclear cells (MNCs).** Human Mo were isolated and cultured as described previously[79]. Briefly, after isolation of blood mononuclear cells (MNCs) by Ficoll–Paque density gradient centrifugation, Mo were isolated using the Pan Monocyte Isolation Kit (Miltenyi Biotec). Purity of isolated Mo was > 90 % as controlled by flow cytometry. Mo were cultured at a concentration of 1 ×10[6] cells/ml in Teflon bags in monocyte medium (McCoy's modified medium (Biochrom AG) supplemented with 1% glutamine, 1% penicillin-streptomycin and 15% fetal bovine serum (FBS)). For SCE-NITH, MNCs were incubated for indicated time periods with 1 μg/ml of phenol-extracted LPS (*Escherichia coli* O55:B5; Sigma-Aldrich) that highly specifically targets TLR4[80]. For gene expression studies, Mo were left untreated or incubated for 4 h with 10 ng/ml LPS. For cytokine studies, Mo were left untreated or incubated for 16 h with 100 ng/ml LPS. In LPS pretreatment experiments, Mo were low-dose pretreated with 0.1 ng/ml LPS for 16 h followed by activation with 1 μg/ml LPS for 4 h. This strategy ensures a significant time of continuous LPS exposure inducing Mo reprogramming[79] while still allowing subsequent LPS activation for 4 h within the lifespan of isolated Mo[81]. To assess primary and secondary LPS responses while glycolysis or OXPHOS were blocked, untreated and LPS-pretreated Mo were stimulated for 4 h with 1 μg/ml LPS while cultivated in Teflon bags in supplemented monocyte medium without and with the addition of either 100 mM 2-Deoxy-D-Glucose (DG) (ThermoFisher) or 1 μM oligomycin A (Santa Cruz). For TF inhibition experiments, AD MNCs were incubated for 4 h with either 10 μM HLM006474 (HLM) (MedChem-Express, Cat# HY-16667), an inhibitor of *E2F1*[30,31], 1 μM all-trans retinoic acid (ATRA) (Merck, Cat# R2625), an inhibitor of *MYB*[32,33], 1 μM

fludarabine (FA) (MedChemExpress, Cat# HY-B0069), an inhibitor of *STAT1*[34,35], or 100 nM camptothecin (CPT) (MedChemExpress, Cat# HY-16560), an inhibitor of *FLII*[36,37], and subjected to SCENITH studies.

**Human monocyte-derived macrophages (MDM).** Isolated AD and cord blood Mo were seeded in RPMI 1640 medium (Biochrom AG) supplemented with 10% of a heat-inactivated human AD blood plasma pool, 1% glutamine and 1% penicillin-streptomycin at a concentration of $2 \times 10^6$ cells/ml and cultivated up to 14 days for differentiation into MDM. In selected experiments, the MDM medium was supplemented with 10% FBS and 50 ng/ml GM-CSF and 50 ng/ml M-CSF (both Peprotech) instead of AD blood plasma pool or pooled blood plasma from adult and new-born (cord blood) blood donors was used in an autologous and cross-heterologous manner for MDM differentiation. In TF inhibition experiments, the AD plasma-based MDM medium was supplemented with either 10 μM HLM, 1 μM ATRA, 1 μM FA, or 100 nM CPT during the first 2 days of cultivation. Every third day 30% of the medium was replaced by fresh medium. After indicated time periods, MDM numbers were determined by manual microscopical counting of 4 high-power fields (HPF) and extrapolation to the total culture well. Subsequently, cells were harvested for flow cytometric analyses. For May-Grunwald-Giemsa (MGG, Pappenheim) staining air-dried and 2% paraformaldehyde (PFA) fixed MDMs cultured in the same manner in Lap-Tek chambers (ThermoFisher) were used. For the analysis of cellular functions, cultured MDMs were stimulated for 4 h and 16 h with 10 ng/ml LPS or phosphate-buffered saline (PBS) (controls) and culture supernatants were collected for cytokine quantification. The effect of glycolysis and OXPHOS inhibition on MDM differentiation was tested by cultivating seeded Mo for the first 24 h in supplemented (1% glutamine, 1% penicillin-streptomycin, 15% FBS) DMEM standard medium (Gibco; 16.7 mM glucose (G-high)), DMEM low glucose medium (R&D; 5.5 mM glucose (G-low)), and DMEM low glucose with either 100 mM DG or 1 μM oligomycin and subsequent cultivation in standard MDM medium. Cell counting and apoptosis measurements were done in d1 MDM and macrophage marker expression studies in d3 MDM.

**Jurkat cells.** Jurkat T leukemia cells (clone E6–1) were purchased from the American Type Culture Collection (ATCC, Manassas, VA, USA). The Jurkat cells were cultured at 37 °C in RPMI 1640 medium supplemented with 10% FBS and 1% penicillin-streptomycin. Cells were subcultured at a ratio of 1:4 upon reaching a cell density of 70–80%. For FACS proliferation studies cells were used at passages 3 or 4.

### RNA isolation, library construction, RNA sequencing, and real-time quantitative reverse transcription PCR (qRT-PCR)
Total RNA was isolated from human Mo with the NucleoSpin RNA Mini kit (Macherey-Nagel) according to the manufacturer's protocol. For bulk RNA sequencing, cDNA libraries were prepared from 100 ng total RNA using the TruSeq® Stranded Total RNA Library Prep Globin from Illumina. The final libraries were quantified using a real-time PCR with KAPA HiFi HotStart (Roche) and the size distribution was measured using the Agilent high sensitivity D1000 assay on a TapeStation 4200 system (Agilent technologies). Sequencing was performed in single-end mode on a HiSeq1500 (Illumina) with a TruSeq SBS Kit v3-HM (50 cycles) Kit. Base call files were converted to fastq format using bcl2fastq v.2.20. For qRT-PCRs, cDNA was synthesized from 300 ng of total RNA using the

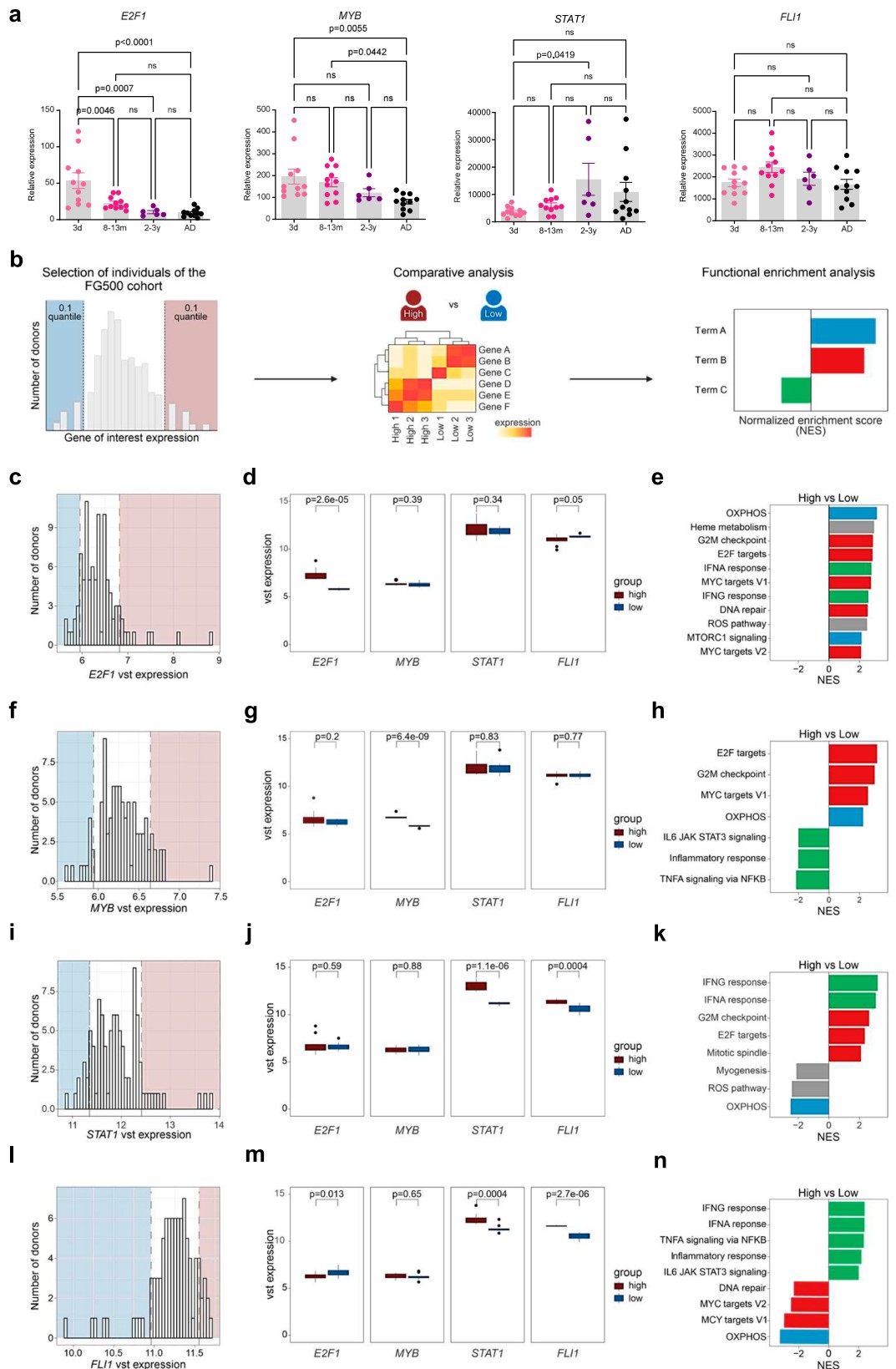

RevertAid Revese Transcriptase Master Mix (ThermoFisher Scientific) following the manufacturer´s recommendations. qRT-PCR was done as described previously[4]. The human primers used for qRT-PCRs were *GAPDH* (F, GCAAATTCCATGGCACCGT; R, GCCCCACTTGATTTTGGA GG), *E2F1* (F, TCGTAGCATTGCAGACCCTG; R, ACATCGATCGGGCCTT

GTTT), *MYB* (F, CGCAGCCATTCAGAGACACT; R, GGTAGCACCTGCTG TCCTTT), *STAT1* (F, TGTATGCCATCCTCGAGAGC; R, AGACATCCTGCC ACCTTGTG), and *FLI1* (F, GGCTGTAACCGGGTCAATGT; R, GTCAAAGA GGGACTGGTCGT). Sample data are presented as target gene expression relative to the housekeeper GAPDH.

**Fig. 7 | Population-based transcriptional analysis of *E2F1*, *MYB*, *STAT1*, and *FLI1* functions. a** Expression of *E2F1*, *MYB*, *STAT1* and *FLI1* in Mo isolated from 3 days old healthy newborns (*n* = 11), 8–13 months old infants (*n* = 11), 2–3 years old toddlers (*n* = 6), and adults (AD, *n* = 11) plotted as means ± SEM. p-values were determined using one-way ANOVA and *post hoc* Tukey's multiple comparison tests. **b** Concept of human variation analysis using transcriptome data of donors of the FG500 whole blood dataset. **c**, **f**, **i**, **l** Donors are binned by gene expression and the top and bottom 10% of donors represent groups with high and low expression of indicated genes of interest. **d**, **g**, **j**, **m** Box plots showing the variance stabilized mean expression (central line) and interquartile ranges (box edges) of *E2F1*, *MYB*, *STAT1* and *FLI1* in respective *E2F1* (**d**), *MYB* (**g**), *STAT1* (**j**) and *FLI1* (**m**) high and low groups (each *n* = 10). Whiskers extend to the smallest and largest data points within 1.5 times the interquartile ranges. Points outside the whiskers are outliers. p-values were determined using unpaired two-sided t-tests. **e**, **h**, **k**, **n** GSEA using Hallmark terms based on the ranked gene list of the comparison of *E2F1* (**e**), *MYB* (**h**), *STAT1* (**k**) and *FLI1* (**n**) high *versus* low individuals. Results are filtered for an absolute normalized enrichment score (NES) > 2. Bars are colored by main functional categories. Source data are provided as a Source Data file.

### Bioinformatic analysis of transcriptomic data

**Pre-processing of transcriptomic data.** Sequenced reads were randomly downsampled to 10 million reads per sample and remaining reads were aligned and quantified using STAR: ultrafast universal RNA-seq aligner (v2.7.3a)[82] and the human reference genome GRCh38p13 from the Genome Reference Consortium.

For bioinformatics analyses, raw counts were imported into R using the R docker image jss/jss_R412_S41. Genes with less than 10 reads in 3 or more samples, as well as the mitochondrial genes MT-RNR1 and MT-RNR2, were excluded. The R/DESeq2 package (v1.34.0)[83] was used for gene count normalization and rlog transformation. Batch correction was performed using surrogate variable analysis (SVA) from the R/sva package (v3.42.0)[84]. The R/limma package (v3.50.1)[85] was used for the calculation of batch-corrected counts by correcting for SV1-3. Differentially expressed genes were calculated following the DESeq2 pipeline with multiple-testing correction by independent hypothesis weighting from the R/IHW package (v1.22.0)[86], an adjusted p-value cutoff of 0.05 and a fold-change (FC) cutoff of 2. FC shrinkage using apeglm (v1.16.0)[87] was applied.

**hCoCena - Horizontal Construction of Co-expression network analysis.** To elucidate similarities and differences within the gene expression patterns of NEO and AD Mo at baseline and following LPS activation, co-expression network analysis was performed using the R/hCoCena package (v1.0.0)[88] and the R docker image mo126/hcocena. To identify genes with similar expression patterns across samples, pairwise Pearson correlations were calculated on the batch-corrected and log-transformed expression values of the 8535 most variable genes, selected based on the result of the *suggest_topvar* function. Data were filtered for correlation values > 0.963 to maintain a network with scale-free topology ($R^2$ = 0.854) consisting of 6517 nodes and 184361 edges. The group fold change (GFC) was calculated for each gene and each condition on the inverse logarithmic count data. Unbiased clustering was performed using the Leiden algorithm. Network visualization was performed using the Organic Layout in Cytoscape (v3.8.2). Clusters with less than 40 genes are not shown.

Gene set enrichment analysis (GSEA) was performed on the age- and treatment group-related modules using the built-in functions. Information on hallmark terms was obtained from the gene set file "h.all.v7.4.symbols.gmt" downloaded from the Molecular Signatures Database (MSigDB).

Hub gene detection based on weighted degree centrality, weighted closeness centrality and weighted betweenness centrality was performed for selected modules. Results were filtered for the 15 highest-ranked transcription factors.

**Transcription factor binding site (TFBS) motif enrichment analysis.** TFBS motif enrichment analysis was performed using the R/RCisTarget package (v1.14.0)[89] based on the gene-motif rankings database with a search space of 10 kb up- and downstream of the transcriptional start site ("hg38_refseq-r80_10kb_up_and_down_tss.mc9nr_.feather") and the human motif annotation database "motifAnnotations_hgnc" integrated in RCisTarget. Results were filtered for transcription factors annotated with high confidence and a normalized enrichment score (NES) > 3.

Networks of selected transcription factors, as well as their target transcription factors, also showing enriched binding motifs among the gene lists of interest, were constructed using the R/network package (v1.18.0)[90] and the R/ggnetwork package (v0.5.10)[91] with the "fruchtermanreingold" layout.

**Human variation analysis (HUVA).** The R/huva package (v0.1.5)[28] was used based on the "FG500_whole_blood" dataset from the R/huva.db package (v0.1.5) using the R docker image bonaguro/huva_docker:015. Individuals were ranked based on their expression of the gene of interest, and individuals within the top and bottom 0.1 quantiles were selected for further analyses (n = 10 per group). GSEA was performed using default settings based on the ranked gene lists in the comparison of high *versus* (vs) low groups. Results were filtered for an absolute normalized enrichment score (NES) > 2.

### Stable isotope tracing, GC−MS measurement and data analysis

The tracer medium was prepared in SILAC RPMI 1640 Flex Media without glucose and glutamine (Thermo Fisher Scientific) plus L-arginine, L-lysine, non-labelled glutamine (2 mM; Biochrom AG) and either d-glucose-$^{13}C_6$ (11.1 mM; Sigma-Aldrich) or non-labelled d-glucose-$^{12}C$ (11.1 mM; Roth). Isolated Mo were cultured in Teflon bags in tracer medium for 24 h to reach isotopic steady state. Metabolite extraction was performed as previously described (Bambouskova et al., 2021; Sapcariu et al., 2014). Briefly, Mo were washed with 0.9% NaCl and then quenched with pre-cooled HPLC-grade methanol (− 20 °C, 0.2 ml per bag). After adding an equal volume of 4 °C deionized water containing 1 μg/ml $D_6$-pentanedioic acid (C/D/N isotopes) as internal standard, cells were transferred to microcentrifuge tubes pre-added with 0.25 ml of pre-cooled (− 20 °C) HPLC-grade chloroform. The extracts were vortexed at 1400 r.p.m. for 20 min at 4 °C and centrifuged at 17,000 x g for 5 min at 4 °C. Next, 0.3 ml of the upper aqueous phase was transferred into glass vials compatible with gas chromatography and then dried under vacuum at 4 °C in the CentriVap Concentration System (Labconco). Gas chromatography−mass spectrometry (GC−MS) measurement of relative metabolite levels and isotopic enrichment was performed as described (Sapcariu et al., 2014), using an Agilent 7890B gas chromatograph equipped with a 30-m DB-35ms and a 5-m Duraguard capillary column (Agilent) for separation of derivatized metabolites, and an Agilent 5977B MSD system (Agilent) for measurement of metabolites. Briefly, dried metabolite extracts were derivatized with equal amounts of methoxylamine (20 mg/ml in pyridine) and MTBSTFA before injection into the GC−MS system. Measurements were carried out in either full scan mode or selected ion mode. Processing of chromatograms and the calculation of mass isotopomer distributions as well as relative quantification of metabolites were performed using the Metabolite Detector software (Hiller et al., 2009).

### SCENITH

Human MNCs were cultivated at 37 °C, 5% $CO_2$ in supplemented RPMI medium without streptomycin and treated for 15 min with DG (100 mM), oligomycin A (1 μM) or both (DGO setting) or left untreated (control). Then, puromycin (10 μg/ml; Roth) was added for 15 min at 37 °C. Cells were stained with eBioscience Fixable Viability Dye eFluor

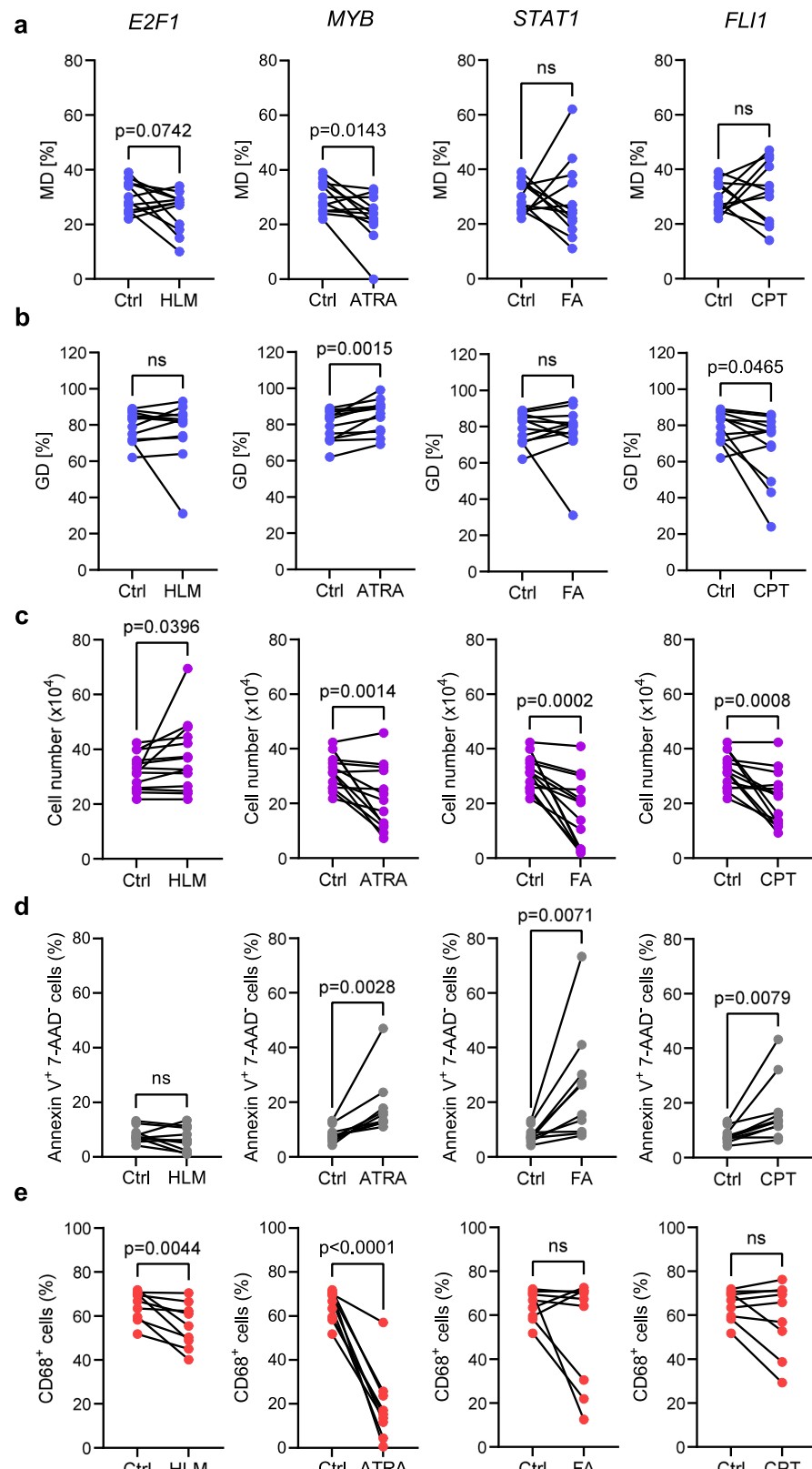

**Fig. 8 | *E2F1*, *MYB*, *STAT1* and *FLI1* are central transcriptional regulators of the cell metabolism and myeloid differentiation in Mo. a**, **b** Isolated AD MNCs were left untreated or pretreated for 4 h with the inhibitors of *E2F1* (HLM), *MYB* (ATRA), *STAT1* (FA) and *FLI1* (CPT) and subjected to SCENITH studies for metabolic profiling of Mo. **a** OXPHOS dependence (MD) and (**b**) glycolytic dependence (GD) of pretreated compared to untreated Mo (Ctrl) (each *n* = 12), respectively. **c**–**e** AD Mo were differentiated for 7 days into MDM without (Ctrl) or in the presence of HLM,

ATRA, FA or CPT during the first 2 days of cultivation. **c** Numbers of Ctrl MDM and inhibitor-treated MDM at harvest on d7 (each *n* = 12). **d** Proportion of apoptotic cells (Annexin V+7-AAD-) in Ctrl MDM and inhibitor-treated MDM (each *n* = 10). **e** Proportion of CD68+ MDM in Ctrl MDM and inhibitor-treated MDM (each *n* = 9). *p*-values were determined using paired one-sided t-tests. Source data are provided as a Source Data file.

506 (1:1000; Invitrogen) in FACS buffer (PBS with 2 mM EDTA and 2% BSA) for 30 minutes at 4 °C and subsequently fixed with 2% PFA. Intracellular staining of puromycin was performed by incubation with the Alexa Fluor 647-conjugated anti-puro monoclonal antibody (1:1000, Clone MABE343-AF647, Sigma-Aldrich) in intracellular staining buffer (FACS buffer with 0.5% saponin and 0.2% Tween20) for 30 min at 4 °C. This protocol was adapted from the original SCENITH kit (http://www.scenith.com) and protocols developed by R. Argüello[11]. Cells were washed twice with FACS buffer and analyzed using a FACS Canto II flow cytometer (BD Biosciences). Data were analyzed using DIVA software v8.0.1 (BD Biosciences) and Kaluza software v2.1 (Beckman Coulter).

### Flow cytometry
Apoptosis and necrosis rates were measured in day 7 MDMs by flow cytometric staining using the APC Annexin V Apoptosis Detection Kit with 7-AAD (Biolegend) according to the manufacturer's protocol. Proliferation activity of human Mo after 24 h of ex vivo cultivation was determined flow cytometrically after intracellular staining with PE mouse anti-human Ki-67 mAb (clone Ki-67, Biolegend). Jurkat cells served as positive controls in this assay. For flow cytometric study of MDM differentiation, marker staining was performed with FITC mouse anti-human CD68 mAb (clone Y1/82 A, Biolegend), PE mouse anti-human CD80 mAb (clone 2D10, Biolegend) and APC mouse anti-human CCR7 mAb (CD197; clone G043H7, Biolegend).

### Cytokine quantification
For measurements of indicated cytokines in cell culture supernatants, we used human LEGENDplex assays (BioLegend). FACS Canto II flow cytometer was used for measurements and the cloud-based LEGENDplex Data Analysis Software Suite Version 2022-07-15 (BioLegend, Qognit) was used for data analysis.

### Statistics
Statistical tests applied for RNA-sequencing data analysis are described above. The statistical significance of two-group comparisons was calculated using Mann-Whitney U (MWU) tests. ANOVA analyses were performed across age and treatment groups. *Post hoc* Tukey's multiple comparison tests were applied between subgroups. To assess the linear relationship between metabolic parameters and cytokine levels, Pearson's correlation coefficient r was computed and the trend line was visualized by linear regression.

A generalized linear model (glm; R stats package v4.2.2) with a Poisson distribution was generated to assess how the variance in energy metabolism, OXPHOS, and glycolytic activity could be explained by selected meta data variables. We tested the interaction effect between the respective metabolic parameters and plotted the standardized estimates (effect sizes) of our model using the *plot_model* function of the R package sjplot (v2.8.3; URL:https://CRAN.R-project.org/package=sjPlot) with default parameters with transform = NULL, active sorting of estimates and p.adjust = "BH". Statistically significant p-values are indicated in the figures, the statistical tests used are specified in the figure legends.

### Reporting summary
Further information on research design is available in the Nature Portfolio Reporting Summary linked to this article.

## Data availability
The raw RNA-seq files are deposited in the EGA database under the accession number EGAS00001007555. The mass spectrometry raw data are deposited on LeoPARD, a repository of the Technical University Braunschweig, under https://doi.org/10.24355/dbbs.084-202312071717-0. All other data are available in the article and its Supplementary files or from the corresponding author upon request. Source data are provided with this paper.

## Code availability
The code necessary to reproduce the analysis is deposited on GitHub under https://github.com/LisaHolsten/Neonatal_monocyte_immunometabolism and Zenodo under https://doi.org/10.5281/zenodo.14886497.

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

## Acknowledgements

This work was supported by grants from the Federal Ministry of Education and Research (BMBF) to SP, CH and DV (PROSPER; 01EK2103B and 01EK2103A, respectively), and the Deutsche Forschungsgemeinschaft (DFG, German Research Foundation) to SP (PI 1512/1-3), DV (VI 538/6-3 and VI 538-9-1) and TU (UL 521/1-1). Further support was provided by the DFG SFB 1583/1 ("DECIDE") project number 492620490 to WK, MV and DV, the DFG TRR 359 ("PILOT") project number 491676693 to DV, the DFG Germany's Excellence Strategy – EXC 2155 'RESIST' – Project ID 390874280 to GH and DV. MB received funds from the HGF Helmholtz AI grant Pro-Gene-Gen (ZT-I-PF5-23) and BMBF PriSyn (16KISA030). JLS received funds from the BMBF-funded excellence project Diet-Body-Brain (DietBB) under grant number 01EA1809A, the DFG (projects numbers 347286815, 329123747, and 423957469), Germany's Excellence Strategy (EXC2151-390873048, ImmunoSensation2), the Helmholtz-funded consortium Sparse2Big, and the Joint Programming Initiative, A Healthy Diet for a Healthy Life (JPI HDHL; project 529051018). We are grateful to all infants and children and their parents and all adult volunteers who participated in our study. We thank Heidi Theis and Michael Kraut for their great technical assistance in the preparation of libraries for RNA sequencing.

## Author contributions

G.E., A.M.T., L.H., T.U., and D.V. contributed to conception, design and methodology of the study. G.E., A.M.T., M.W., J.H., J.S., M.R., A.S.H., A.H., K.Hä, K.Hi, M.V. performed experiments. S.P., C.D., J.B., A.W., Cv.K., and G.H. supported the recruitment of study participants and biosampling. LH, MB, and TU performed computational studies. L.H., C.H., T.U., and DV wrote the first draft of the manuscript. G.E., A.M.T., M.W., S.P., A.H., K.Hi, J.L.S., W.K., and M.V. contributed to manuscript revision and editing. S.P., G.H., C.H., T.U. and D.V. acquired funding. All authors read and approved the submitted version.

## Funding

## Competing interests

The authors declare no competing interests.

## Additional information

[1]Department of Pediatric Pneumology, Allergology and Neonatology, Hannover Medical School, Hannover, Germany. [2]Department of Pediatrics, University Hospital Würzburg, Würzburg, Germany. [3]Genomics and Immunoregulation, Life & Medical Sciences (LIMES) Institute, University of Bonn, Bonn, Germany. [4]Systems Medicine, German Center for Neurodegenerative Diseases (DZNE), Bonn, Germany. [5]PRECISE Platform for Single Cell Genomics and Epigenomics, DZNE and University of Bonn, Bonn, Germany. [6]Department for Bioinformatics and Biochemistry, BRICS, Technical University Braunschweig, Braunschweig, Germany. [7]Institute of Human Genetics, University of Lübeck, Lübeck, Germany. [8]Modular High Performance Computing and Artificial Intelligence, Deutsches Zentrum für Neurodegenerative Erkrankungen (DZNE), Bonn, Germany. [9]Department of Gynecology and Obstetrics, University Hospital Würzburg, Würzburg, Germany. [10]Department of Obstetrics and Gynecology, Hannover Medical School, Hannover, Germany. [11]Cluster of Excellence RESIST (EXC 2155), Hannover Medical School, Hannover, Germany. [12]Würzburg Institute of Systems Immunology, Max Planck Research Group, Julius-Maximilians-University Würzburg, Würzburg, Germany. [13]Center for Infection Research, University Würzburg, Würzburg, Germany. [14]These authors contributed equally: Greta Ehlers, Annika Marie Tödtmann, Lisa Holsten. ✉e-mail: viemann_d@ukw.de

