## [Transparent Peer Review file · Nature Communications]

Oxidative phosphorylation is a key feature of neonatal monocyte immunometabolism promoting myeloid differentiation after birth

Corresponding Author: Professor Dorothee Viemann

Version 0:

Reviewer comments:

Reviewer #1

(Remarks to the Author)

The authors provide a well-written and scientific sounds manuscript relevant in the field of neonatal and pediatric immunity. The authors provide a broad sample set, making the study interesting in the sense of immune ontogeny. The methods utilized are complementary and described in sufficient details. In some sections, the rationale why the authors used certain approach are not clear.

Given the journal's broad readership, I suggest indicating specific age ranges for the used participants in the introductory text.

In the first results section, the authors have used cord-blood-derived monocytes. This should be stated more clearly in the text, as merely mentioning 'healthy newborns' is not sufficiently descriptive.

Considering recent research (DOI: 10.1016/j.molcel.2022.11.017), cell cycle control appears to be a crucial factor in macrophage polarization. Could this also play a role in the differing responses at the monocyte level?

For Figure 2B, I recommend adding scales to the plots to improve clarity and comprehension.

Regarding the lipopolysaccharide used, was it specific to TLR4, or did it also target TLR2? The term "microbial training" used in the manuscript is somewhat misleading. The field commonly uses the term "trained innate immunity," so I suggest using "LPS-pretreatment" for clarity.

The rationale behind using a 16-hour stimulation period for LPS pre-stimulation is not clear. Could the authors elaborate on this choice?

The manuscript states that all infants involved were breastfed. It's unclear whether this refers to the time of sampling or if it implies that they were exclusively breastfed for the first six months. Clarification on whether these infants were exclusively or partially breastfed is necessary.

The authors have to critically rethink their statement: "Breast milk is a high-fat and slightly carbohydrate-restricted diet (Supplementary Table 4) and hence a KD". Breast milk alone is not a therapeutic "ketogenic diet". The cited study showed that breast feeding or expressed breast milk is possible besides ketogenic formula. You will never reach the desired level of therapeutic ketosis with mother milk alone in infants with epilepsy. Therefore, this comparison is critical and the authors might need another citation to proof their point. The authors took blood from the babies. Did the measure the level of ketosis within the samples to underline their theory? Are there any data concerning ketosis levels in healthy infants with 3- 6 months?

The selection of a seven-day ketosis period in adults raises questions. Is this duration based on the lifespan of circulating monocytes? Is this sufficient to see metabolic changes which might be linked also to epigenetic changes? The authors should provide their rationale in the manuscript and discuss their limitations.

The authors' choice of differentiating monocyte-derived macrophages (MDMs) without cytokine-driven stimuli is intriguing. Could there be a difference if the monocytes were differentiated using GM-CSF/M-CSF stimuli? This point warrants further discussion.

The manuscript should include dot plots for the flow cytometry experiments to offer a more detailed data representation.

The study's reliance on bioinformatic inferences based on transcription factors without experimental validation is a notable limitation. While the data might indicate these factors' involvement, definitive validation is absent. It's acceptable if this is clearly stated as a limitation in the discussion section. However, whether this is sufficient for publication should be determined by the editor.

Reviewer #2

(Remarks to the Author)

In this study, the authors investigated age-dependent changes in innate immune cell metabolism and function. They performed experiments on up to 4 different age groups (newborn, infant, toddler, adult). Through multiple lines of evidence (transcriptomic, isotope tracing and metabolic data) they show that neonatal M₀s are characterized by ongoing cell differentiation linked to a high OXPHOS activity and low glycolytic activity. With increasing age, this balance shifts towards lower OXPHOS activity and higher glycolytic activity, the latter of which correlates with inflammatory responsiveness (primarily using TNF production as a proxy).

This is an interesting concept and the authors are commended for assembling samples from a variety of age groups, some of whom are very tedious and hard to attain. In addition, this topic is very important as it has implications for understanding neonatal immunometabolic responses to infection, vaccination and development of chronic disease, and potentially developing interventions to perturb implicated pathways for clinical benefit.

Below are my comments to improve this important manuscript:

-The dose of a microbial stimulus can influence the direction of the immune response in various ways. For example, it is well-known that high doses of a microbial stimulus may result in immune exhaustion or paralysis, while lower doses may result in a pro-inflammatory phenotype. According to the Methods, the authors used variable doses of LPS for different experiments. It would be helpful for the reader if the authors a) indicated the dose and duration of LPS treatment in each figure legend, and b) provided justification as to why they chose different LPS doses. I would be keen to see if there are age-specific responses using the same dose of LPS across experiments.

-The terms metabolic reprogramming and immune training are often used in the text to describe some phenomena observed by the authors that do not necessarily reflect these processes, as defined in the literature, also cited in the references of this manuscript. Innate immune training is characterized by long-term epigenetic and metabolic reprogramming of innate immune cells, which manifest days to weeks after an initial stimulation and are induced by a secondary unrelated stimulus after the cell has returned to its pre-stimulated state. This is different from the acute changes observed shortly after LPS stimulation as performed in this paper.

-lines 211-213: The authors employ an interesting experimental design based on a ketogenic diet intervention to determine if dietary factors can explain immunometabolic differences between NEO and AD. However, the comparison of breast-milk to a ketogenic diet is inaccurate. Approximately 39% of breastmilk consists of carbs which makes it a moderately high carb content diet. Also, breastmilk composition changes rapidly during the first few months of life, which is when major immune maturation takes place.

In addition, ketosis in newborns does not occur due to their diet, but is physiologic due to their developmental needs: ketones are a precursor to important metabolites that help with brain development, including cholesterol and amino acids.

-lines 239-240:

There were higher rates of NEO M₀ death compared to AD. According to the Methods, those cultures were supplemented with 10% heat-inactivated human AD blood plasma, implying that no age-specific plasma was used for each age group. There is plenty of literature emphasizing the importance of using age-specific plasma in human in vitro assays (Alipour, R., Fatemi, A., Alsahebhosul, F. et al. Autologous plasma versus fetal calf serum as a supplement for the culture of neutrophils. *BMC Res Notes* 13, 39 (2020). <https://doi.org/10.1186/s13104-020-4902-z>; Angelidou A, et al. Human Newborn Monocytes Demonstrate Distinct BCG-Induced Primary and Trained Innate Cytokine Production and Metabolic Activation In Vitro. *Front Immunol.* 2021 Jul 13;12:674334. doi: 10.3389/fimmu.2021.674334.), especially since neonatal plasma is particularly rich in age-specific immunomodulatory factors (Sanchez-Schmitz, G. & Levy, O. Development of newborn and infant vaccines. *Sci. Transl. Med.* 3, 9027. <https://doi.org/10.1126/scitranslmed.3001880> (2011).) and other microphysiologic components that can influence cellular well-being, immune responsiveness and viability.

-Are there any paired samples between experiments or were all assays run in different datasets? Given the wide inter-donor variability in the readouts tested, running more than 1 assay in the same donor can strengthen the credibility of the results, especially for experiments where the N is low.

Figure 1:

-panels e and f: I guess the results plotted reflect the LPS-treated conditions. Can the authors also plot the LPS-untreated controls? For panels h-l, it is unclear if the results plotted reflect the LPS-stimulated conditions or quiescent cellular states.

Please clarify.

Figure 2:

-SCENITH: were these studies performed on MNCs or Mo? Please be precise (figure legend states Mo, panel a states MNC, methods state MNC). MNC includes cell populations other than Mo.

-panel g-j: why not plot all age groups? Can the authors use different shapes to show which data points belong to each age group? Alternatively, can the authors plot the cytokine data for each age group?

-panel k: how were the clinical data variables chosen? The number of vaccinations or vaccines does not apply to the newborn group, who presumably is not yet vaccinated (especially if cord blood was used). Also, it is a huge leap to assume that high number of siblings is associated with metabolic reprogramming because of higher microbial exposure. Also, the effect size for this observation is really small.

Figure 3: Microbial education induces a metabolic shift towards an AD-like phenotype:

-I am not sure what the authors mean by microbial "education". What they performed is microbial stimulation with LPS (please clarify dose). The relevant comparisons here are LPS vs control, not LPS 4h vs 24h, which indicates the kinetics of the response [unless I misunderstood the design and the LPS 4h and 24h refer to pretreatment for 4h and re-stimulation for 24h]. My interpretation of the data shown is that in NEO Mo, LPS stimulation decreased mitochondrial dependence over the control (baseline) but had no effect on glycolytic dependence (since change from baseline was not significant). In the AD Mo, LPS stimulation had no effect on mitochondrial dependence compared to control, but significantly increased glycolytic dependence (though the effect size seems quite small). Therefore, I find the summary in page 9, line 191-194 inaccurate. Since NEO and AD have different baseline metabolic needs, comparing NEO and AD (such as is done in Suppl figure 3) is meaningful only if their baseline is somehow corrected for/accounted for.

-panels c-e: These experiments were reportedly done to test whether the LPS-induced metabolic shift in NEO Mos also comes with a functional maturation towards an "AD-like immune phenotype". What is the hypothesized AD functional phenotype and in which state are the authors referring to (LPS-treated, LPS-pretreated)? Why is pre-treatment with LPS required to show changes in the functional maturation of Mo? Flow cytometry for cellular differentiation/maturation markers in LPS-treated Mo would have been more informative. These panels are distracting and the message is unclear. It would be more helpful to show cytokine/chemokine concentrations after primary and secondary stimulation to determine changes compared to the immediately pre-existing cellular state. In lines 197-202, the authors report that "LPS pre-treatment followed by LPS restimulation promoted a stronger induction of TNF α in NEO Mo corroborating our finding that LPS exposure promotes glycolytic activity on NEO Mos". Do they have metabolic data on the same samples that were pre-treated and restimulated with LPS?

It is very likely that pre-treatment and re-stimulation could influence metabolic shifts differently than a single microbial stimulation. If those experiments were not done, this claim cannot be made (also, as mentioned above, there was non-significant effect on NEO glycolysis in panel b). One way to address this question would be to block glycolysis and measure the effects on cytokine secretion.

Reviewer #3

(Remarks to the Author)

In the manuscript entitled "Oxidative phosphorylation is a key feature of neonatal monocyte immunometabolism promoting myeloid differentiation after birth" Ehlers and coworkers combine transcriptomic, metabolic, and immunological approaches in monocytes from healthy individuals at different ages to unravel the ontogenetic differences in the metabolic features and their relationship with myeloid differentiation and inflammatory response. In general this study is very relevant to the field, as it contributes to the characterization of human macrophages and their metabolic features with respect to age and, as such, represents an important step toward the ontogenetic knowledge of human macrophages metabolism and function, which is currently quite limited if compared to murine macrophages. For this reason I believe that this work fills a gap in our knowledge of human macrophages. In particular, the authors are presenting a cutting edge approach to unravel age-dependent differences in human monocytes and macrophages.

However considering the methods applied in this study, I believe that this manuscript would fit much better a more specific system biology journal. The conclusions on the dependency of NEO M0 on OXPHOS are not validated and demonstrated only limitedly at a molecular level. The mechanisms underlying the relationship between the transcriptional programs being activated in NEO vs AD macrophages and the corresponding metabolic outcomes are not demonstrated. This might require to incorporate a wide amount of additional new data.

Version 1:

Reviewer comments:

Reviewer #1

(Remarks to the Author)

All raised concerns were adequately addressed in the responses to the reviewers. The authors provide additional valuable experiments underlying their bioinformatic analysis. Congratulations to the authors.

Reviewer #2

(Remarks to the Author)

I very much appreciate the efforts of the authors to address reviewer feedback. A general comment is that the authors vary the experimental conditions (e.g. duration of stimulation and dose of stimulus) and cell types (Mo, MNC, MDM) used throughout the manuscript. Although they have now added rationale behind these choices, interpretation of findings can be challenging.

Some additional specific comments are below:

-Supplementary figure 6 shows panels of cell numbers and percentages of apoptotic and CD68+ cells. The data show that (high) non-physiologic concentrations of growth factors promote cellular growth and differentiation compared to physiologic concentrations of growth factors contained in human plasma. The authors also show new culture data using AD blood plasma for AD cells and NEO blood plasma for NEO cells (unclear if NEO blood plasma is pooled or not), and show better results when using same-age human plasma (NEO_NEO-PL and AD_AD-PL) vs different age (NEO_AD-PL and AD_NEO-PL) human plasma, meaning better cell numbers (d, better though not significantly different by age), lower % of apoptotic cells (e, significant for NEO only) and comparable number of CD68+ cells (f). Stats between NEO_NEO-PL and AD_AD-PL are missing, but in my opinion, using plasma from the same age group (pooled, if not from the same donor) is the gold standard, especially if the authors are trying to show relevance of their ex vivo model for in vivo human biology. Differences between same age vs different age plasma in their effects on cellular differentiation and effector functions (vs just cell numbers) are not shown.

In their rebuttal the authors state that they were “specifically interested in the cell-intrinsic immunometabolic programming of neonatal and adult monocytes and related capacity to differentiate into MDM. Varying both parameters, cells and plasma, at once makes it difficult ascribing differential outcomes to either the cellular programming or the plasma composition. It was therefore important for us to use the same standardized and well-defined culture medium condition”. It is well known that the cellular microenvironment (in this case the culture medium and any added factors) affects “cell-intrinsic” responses to a great degree. The peculiarities of neonatal immune responses are largely attributable to the unique composition of their plasma and dynamic changes thereof, which is very different from that of adults. The use of plasma from a different age group could obscure soluble plasma-based ontogenic differences that shape cellular immune responses (Pettengill et al, Soluble mediators regulating immunity in early life, Front Immunol 2014). Although the authors express concern about varying 2 parameters (cells and plasma), I would argue that those 2 parameters are interdependent and should be paired by age (NEO cells in NEO plasma and AD cells in AD plasma), especially if the work describes age-specific effects on metabolism. This is a major limitation in the model that should be highlighted in the discussion.

#6: In response to authors’ response:... We added the LPS-untreated controls to Fig. 1 e,f. After 4h of cultivation, there was no significant cytokine accumulation detectable in both NEO and AD monocyte cultures.

From what I can see and what is noted in the text, there IS significant cytokine accumulation in both NEO and AD monocyte cultures: “Higher LPS-induced TNF- α production by AD Mo compared to NEO could also be validated at the protein level (Fig. 1e)” and “NEO Mo distinguish from AD Mo by a higher GM-CSF production (Fig. 1f)”

#8: In response to authors’ response: ...we have changed the wording related to Fig. 3b (now: “LPS stimulation increased glycolysis in NEO Mo only slightly with strong inter-individual variance but in AD Mo uniformly and highly significant (Fig. 3b).”) and moderated the summary (now: “Summarized, these studies suggested that a short-term single LPS treatment induced a metabolic reprogramming of NEO Mo towards an AD-like baseline phenotype, particularly by decreasing OXPHOS and slightly increasing glycolysis.”).

I believe Fig 3a-b have tested MNCs not Mos. If so, please correct.

-Figure 2 g-j: consider different regression lines for NEO and AD as the trends appear to differ by age.

-Lines 456-459: “In our study, LPS pretreatment had age-dependently opposite effects and enhanced subsequent LPS responses in NEO Mo but paralyzed those of AD Mo, suggesting that the initial state of metabolic programming might determine the educational outcome.” The use of the phrase “educational outcome” hints at trained immunity. Although this work may have implications for innate immune training, the authors did not model monocytic trained immunity here, as they only cultured Mos for 24-48h. Please rephrase.

Reviewer #3

(Remarks to the Author)

Considering the amount of additional data provided by the authors, the present version of the manuscript is significantly improved and can be accepted for publication.

Point-by-point response to reviewers

We highly thank all reviewers for taking the time to revise our work, and for the valuable input to substantially improve the manuscript. According to the comments below, we have revised the manuscript and highlighted all changes to the text in yellow.

List of new data generated in additional experiments:

- **Figure 3f-i** illustrate the link between cell metabolism and inflammatory responsivity in monocytes showing that the strong inflammatory response of AD Mo to a primary LPS stimulus and that of NEO Mo to a secondary LPS stimulus after LPS pretreatment is dependent on their glycolytic activity but not their OXPHOS activity.
- **Figure 8a-e** show the impact of pharmacological inhibition of the candidate transcriptional regulators on monocyte's cell metabolism and the outcome of myeloid differentiation. Briefly, the data largely confirm the computationally derived predictions that E2F1 and MYB promote OXPHOS and high-quality myeloid differentiation. Furthermore, as predicted, glycolysis was inhibited by MYB but promoted by FLI1, while MYB, STAT1 and FLI1 in concert promoted cell survival during myeloid differentiation.
- **Supplementary Figure 6a-c** show that using a standard protocol of high-dose GM-CSF/M-CSF-driven MDM polarization instead of our protocol performing MDM differentiation in the presence of AD blood plasma (with physiological concentrations of GM-CSF/M-CSF), maximized the yield of well-differentiated MDMs but concealed the age signature of NEO and AD MDM differentiation.
- **Supplementary Figure 6d-f** show that using cord blood plasma instead of AD blood plasma did not change the outcome of NEO and AD MDM differentiation significantly, however, the differences between NEO MDM and AD MDM became blurry under cord blood plasma, likely due to the higher compositional variability of cord blood compared to healthy adult donor blood.

Reviewer #1:

The authors provide a well-written and scientific sounds manuscript relevant in the field of neonatal and pediatric immunity. The authors provide a broad sample set, making the study interesting in the sense of immune ontogeny. The methods utilized are complementary and described in sufficient details. In some sections, the rationale why the authors used certain approach are not clear.

1. Given the journal's broad readership, I suggest indicating specific age ranges for the used participants in the introductory text.

Answer: We thank the reviewer for this very good suggestion. We followed the advice and added the specific age ranges of the age groups studied in this work to the last paragraph of the Introduction.

2. In the first results section, the authors have used cord-blood-derived monocytes. This should be stated more clearly in the text, as merely mentioning 'healthy newborns' is not sufficiently descriptive.

Answer: According to the reviewer's suggestion we now state the usage of cord blood in the first Results section more clearly as follows. "[...] we performed RNA sequencing of untreated and LPS-activated Mo isolated from cord blood of healthy newborn infants and peripheral blood of healthy adults (Fig. 1a).".

3. Considering recent research (DOI: 10.1016/j.molcel.2022.11.017), cell cycle control appears to be a crucial factor in macrophage polarization. Could this also play a role in the differing responses at the monocyte level?

Answer: This is a very good question of the reviewer. This recent research from Daniel et al. essentially showed that macrophages in proliferation states (S-G2/M-phase) are less sensitive to polarization, i.e. cell differentiation, upon IFN- γ signaling (less plasticity). This is well in line with our findings of a higher differentiation quality of neonatal monocyte-derived macrophages and higher expression of genes involved in cell cycle control, particularly G2/M checkpoint regulators such as E2F1 and MYB. We added this corroborating aspect and paper to the Discussion as follows: "*Recently, macrophages in proliferation states (S-G2/M-phase) have been shown to be less plastic and sensitive to interferon- γ -induced polarization³⁹. This is well in line with our finding of high expression of genes involved in cell cycle control by NEO Mo, particularly G2/M checkpoint regulators, that might warrant high plasticity and high-quality myeloid differentiation.*".

4. For Figure 2B, I recommend adding scales to the plots to improve clarity and comprehension.

Answer: As recommended by the reviewer, we added the scales to the plots in Fig. 2b.

5. Regarding the lipopolysaccharide used, was it specific to TLR4, or did it also target TLR2? The term "microbial training" used in the manuscript is somewhat misleading. The field

commonly uses the term "trained innate immunity," so I suggest using "LPS-pretreatment" for clarity.

Answer: The LPS used in this study has been purchased from Sigma-Aldrich. It is a LPS preparation purified from *Escherichia coli* O55:B5 and additionally phenol-extracted. Phenol extraction has been shown to eliminate contaminants that can activate TLR2-mediated signaling (Hirschfeld et al., 2020, PMID: 10878331). We added this information to the Methods section (paragraph "Cell isolation and culture conditions").

We agree with the reviewer that using "microbial training" might be somewhat misleading as we assess only functional immunometabolic effects of a LPS pretreatment. We followed the reviewer's recommendation and replaced "microbial training" by "LPS-pretreatment" throughout the revised manuscript.

6. The rationale behind using a 16-hour stimulation period for LPS pre-stimulation is not clear. Could the authors elaborate on this choice?

Answer: We thank the reviewer for notifying us that the rationale behind the LPS pretreatment experiments requires clarification.

The concept of microbiome and immune interactions considers continuous low-dose exposure to microbe-associated molecular patterns (MAMPs) from commensal bacteria being essential for maintaining immune homeostasis based on inducing immune tolerance and/or priming the immune system (Rakoff-Nahoum et al., 2004, PMID: 15260992; Foster et al., 2007, PMID: 17538624; Biswas & Lopez-Collazo, 2009, PMID: 19781994; Netea et al., 2016, PMID: 27102489).

Isolated monocytes typically remain viable for about 24 to 48 hours under standard laboratory conditions (Gordon & Taylor, 2005, PMID: 16322748). To achieve continuous low-dose exposure to MAMPs (that starts in neonates with microbial colonization after birth), we chose a 16 hours period of pretreatment with low-dose LPS (0.1 ng/ml). This approach ensures a significant time period of LPS exposure and LPS-mediated reprogramming but still allows performing 4 hours of subsequent high-dose LPS stimulation (1 µg/ml) within the lifespan of isolated monocytes. This strategy has already been successfully employed in our previous work where we used a 16-hour pretreatment period with LPS doses ranging from 0.025–1,000 ng/ml (Austermann et al., 2014).

The group of Mihai Netea pretreats isolated monocytes for 24h with TLR ligands to induce immune training but then supplements the medium with 10% pooled human serum before restimulation after 6 days (Quintin et al., 2012, PMID: 22901542; Cheng et al., 2014; Bekkering et al., 2016, PMID: 24903093; Zhang et al., 2022, PMID: 35133977). Human serum provides survival factors like GM- and M-CSF extending monocyte's viability but also inducing differentiation into macrophages which we wanted to avoid in these specific experiments.

In the revised manuscript, we notify the readership about our rationale by adding the following sentence to the Methods section: *"This strategy ensures a significant time of continuous LPS exposure inducing Mo reprogramming⁷⁶ while still allowing subsequent LPS activation for 4h within the lifespan of isolated Mo⁷⁸."*

7. The manuscript states that all infants involved were breastfed. It's unclear whether this refers to the time of sampling or if it implies that they were exclusively breastfed for the first six

months. Clarification on whether these infants were exclusively or partially breastfed is necessary.

Answer: We are grateful for this important request. We complemented the sentence the reviewer is referring to and in from the readership that *“All infants included in our study were exclusively breastfed for the first six months of life.”*

8. The authors have to critically rethink their statement: “Breast milk is a high-fat and slightly carbohydrate-restricted diet (Supplementary Table 4) and hence a KD”. Breast milk alone is not a therapeutic “ketogenic diet”. The cited study showed that breast feeding or expressed breast milk is possible besides ketogenic formula. You will never reach the desired level of therapeutic ketosis with mother milk alone in infants with epilepsy. Therefore, this comparison is critical and the authors might need another citation to proof their point. The authors took blood from the babies. Did the measure the level of ketosis within the samples to underline their theory? Are there any data concerning ketosis levels in healthy infants with 3- 6 months?

Answer: The reviewer’s point to rethink our statement that breast milk is a sort of ketogenic diet is absolutely valid. We have to admit that this is a clear overstatement and we deleted the following part in this specific sentence: “[...] and hence a KD.”. The reviewer is also right that breast milk alone is not sufficient to achieve therapeutic ketosis wherefore the cited study of Dressler et al. is not optimal.

Our major aim was to introduce the reader to why the difference in diets might explain the striking difference in the cell metabolism between neonates and adults. Several studies showed augmented ketogenesis in breast-fed infants (see below). We did not measure the levels of ketone bodies in the babies of our study population to support these previous studies. However, relating the differences in cell metabolism to exclusive breastfeeding certainly falls too short. To improve the explanation of why we found it worth to test the effect of a KD in adults on cell metabolism, we changed the entire introductory paragraph of the fourth Results section as follows: *“Ketogenesis appears to be an integral part of extrauterine metabolic adaptation in the term human neonate, as from 12 h of age, healthy term infants show high ketone body turnover rates approaching those found in adults after several days of fasting¹⁷⁻¹⁹. All infants included in our study were exclusively breastfed for the first six months of life. Breast milk is a high-fat and slightly carbohydrate-restricted diet (Supplementary Table 4). It has been speculated that augmented ketogenesis in breast-fed infants may be due to the activation of mitochondrial fatty acid β -oxidation by breast milk factors²⁰.”* We also reworded the abstract as follows: “[...] whereas ketogenic diet in adults mimicking neonatal ketosis could not revive a neonate-like metabolism.”.

9. The selection of a seven-day ketosis period in adults raises questions. Is this duration based on the lifespan of circulating monocytes? Is this sufficient to see metabolic changes which might be linked also to epigenetic changes? The authors should provide their rationale in the manuscript and discuss their limitations.

Answer: The rationale of a seven-day KD period in adults was actually less linked to the monocyte lifespan (which is short-lived, typically ranging from 1 to 3 days (van Furth & Sluiter, 1986, PMID: 3944542; Ginhoux & Jung, 2014, PMID: 24854589; Geissmann & Littman, 2003, PMID: 12871640) but rather to the time period of postnatal ketosis. We added our rationale to the fourth Results section as follows: *“One week of KD was chosen since KD in adults induced*

significant ketosis from day 2 on, so that after 7 days of KD ketosis prevailed for at least 5 days (Fig. 4b). This time period corresponded with the time period of elevated ketone body concentrations in neonates during the first 5-7 days of life^{21,22}.

We did not assume that a KD in adults would induce epigenetic changes but eventually a metabolic adaptation of monocytes under ketosis. Whether a long-term therapeutic KD would be able to induce a sustainable reprogramming of cell metabolism that would come with epigenetic changes is an interesting question but beyond the scope of this work. Considering the short lifespan of circulating monocytes, such reprogramming would likely affect the myeloid stem cell niche which would be important to include in the work plan.

With respect to the neonatal cell metabolism, we believe that breast milk feeding might support cell growth and differentiation by fueling OXPHOS whereas both continuous low-dose microbial exposure to commensals and recurrent challenges with pathogens are the major driver of metabolic reprogramming including epigenetic changes.

10. The authors' choice of differentiating monocyte-derived macrophages (MDMs) without cytokine-driven stimuli is intriguing. Could there be a difference if the monocytes were differentiated using GM-CSF/M-CSF stimuli? This point warrants further discussion.

Answer: This is a very good question of the reviewer. To avoid the loss of monocyte age signatures or signatures induced experimentally during *ex vivo* differentiation into macrophages, our group (e.g. Ulas et al., 2019) - similar to the group of Mihai Netea - usually uses pooled human blood plasma of healthy adult volunteers. The physiological plasma concentrations of the macrophage survival and differentiation factors GM-CSF and M-CSF are approximately <1 to 10 pg/ml and 20 to 1000 pg/ml, respectively (Hamilton 2008, PMID: 18551128). In contrast, cytokine-driven polarization of monocytes into macrophages is typically performed at high dosages of 50 ng/ml of GM-CSF/M-CSF, respectively. We always feared that such optimization of medium growth conditions might override the donor-specific monocyte signature. However, we admit that we – and to the best of our knowledge other research groups – have never tested this systematically. Therefore, we agree with the reviewer that it is now time and the opportunity to do it. We performed new experiments to clarify how the differentiation of neonatal and adult monocytes into macrophages changes when using GM-CSF/M-CSF at high concentrations compared to using human blood plasma. The description of the culture conditions has been added to the respective paragraph in the Methods section (*Human monocyte-derived macrophages (MDM)*). The findings confirmed our concerns. They are shown in the new Supplementary Fig. 6a-c and reported in the fifth Results section as follows: “*Of note, using high-dose GM-CSF/M-CSF (50 ng/ml) instead of AD blood plasma (with physiological concentrations of GM-CSF (<1 to 10 pg/ml) and M-CSF (20 to 1000 pg/ml)²⁴, maximized and aligned the yield of well-differentiated MDMs derived from NEO and AD Mo, concealing the age signature of MDM differentiation (Supplementary Fig. 6a-c).*”

11. The manuscript should include dot plots for the flow cytometry experiments to offer a more detailed data representation.

Answer: In the revised manuscript we added a new Supplementary Fig. 5 to include representative plots of the flow cytometry studies in MDMs

12. The study's reliance on bioinformatic inferences based on transcription factors without experimental validation is a notable limitation. While the data might indicate these factors' involvement, definitive validation is absent. It's acceptable if this is clearly stated as a limitation in the discussion section. However, whether this is sufficient for publication should be determined by the editor.

Answer: We agree with the reviewer that experimental validation of the potential role of E2F1, MYB, STAT1 and FLI1 would strongly corroborate our hypothesis. Demonstrating at a population-based level that individuals overexpressing *E2F1*, *MYB*, *STAT1* or *FLI1* show global gene expression patterns that match the predicted functions of these transcription factors (HUVA, Fig. 7b-n) might be too weak for a proof of concept.

Therefore, we first had to find an approach that achieves TF inhibition in primary human monocytes without heavily distorting their primary transcriptional and metabolic programming before the actual experiment starts. We finally decided to employ pharmacological inhibitors and analyzed in a large number of new experiments what effect they had on cell metabolism (SCENITH studies) and MDM differentiation. The respective inhibitors used were:

- HLM006474, an inhibitor of E2F1 (Ma et al., 2008, PMID: 18676853; Shirakawa et al., 2022, PMID: 36198264),
- All-trans retinoic acid (ATRA), an inhibitor of MYB (Mandelbaum et al., 2018, PMID: 30209067; Hanna et al., 2021, PMID: 34091189),
- Fludarabine (FA) an inhibitor of STAT1 (Frank et al., 1999, PMID: 10202937; Jiang et al., 2024, PMID: 38509096), and
- Camptothecin (CPT) as inhibitor of FLI1 (Wang et al., 2021, PMID: 33559345; Schutt et al., 2022, PMID: 36074578).

The detailed description of the experiments has been added to the Methods section. The new data support the bioinformatic predictions and interferences and have been illustrated in the new Fig. 8 and reported in a new last Results paragraph as follows: *“To assess the effect of inhibition of the candidate TFs on predicted functions, we avoided genetic editing of Mo to not distort their primary transcriptional and metabolic programming but decided to employ pharmacological inhibitors. Therefore, AD Mo were treated with either HLM006474 (HLM) inhibiting E2F1^{30,31}, all-trans retinoic acid (ATRA) inhibiting MYB^{32,33}, fludarabine (FA) inhibiting STAT1^{34,35}, and camptothecin (CPT) inhibiting FLI1^{36,37}. Even though these inhibitors might have broader effects than exclusively inhibiting E2F1, MYB, STAT1, and FLI1, respectively, the consequences for metabolic pathways activities and MDM differentiation largely supported the computationally predicted TF functions. Inhibition of E2F1 slightly decreased the OXPHOS activity of Mo (Fig. 8a) while glycolysis remained largely unaffected (Fig. 8b). With respect to the outcome of MDM differentiation, E2F1 inhibition yielded higher numbers of MDMs (Fig. 8c) but lower proportions of well-differentiated CD68 macrophage marker positive MDMs (Fig. 8e), while no effect on apoptosis was observed (Fig. 8d). MYB inhibition led to a significant downregulation of OXPHOS but also activated glycolysis (Fig. 8a,b), which together hampered MDM differentiation profoundly as reflected by low MDM yield, increased apoptosis rates and strongly reduced numbers of CD68-expressing MDMs (Fig. 8c-e). Inhibition of STAT1 had no clear effect on Mo cell metabolism while FLI1 inhibition led to a significant decrease in glycolysis but no perceivable effect on OXPHOS (Fig. 8a,b). However, inhibition of both STAT1 and FLI1 hampered MDM yield due strongly enhanced apoptosis but did not impair the quality of MDM differentiation in terms of proportions of CD68⁺ cells (Fig. 8c-e). Thus, in essence, E2F1 and MYB concordantly promoted OXPHOS and high-quality myeloid differentiation. In contrast, inhibition of MYB, STAT1 and FLI1 promoted cell death*

during myeloid differentiation, which was therefore unlikely linked to glycolysis since glycolysis was inhibited by MYB but promoted by FLI1.” The related section in the Discussion has been slightly adapted (lines 473-495).

Reviewer #2:

In this study, the authors investigated age-dependent changes in innate immune cell metabolism and function. They performed experiments on up to 4 different age groups (newborn, infant, toddler, adult). Through multiple lines of evidence (transcriptomic, isotope tracing and metabolic data) they show that neonatal M₀s are characterized by ongoing cell differentiation linked to a high OXPHOS activity and low glycolytic activity. With increasing age, this balance shifts towards lower OXPHOS activity and higher glycolytic activity, the latter of which correlates with inflammatory responsiveness (primarily using TNF production as a proxy).

This is an interesting concept and the authors are commended for assembling samples from a variety of age groups, some of whom are very tedious and hard to attain. In addition, this topic is very important as it has implications for understanding neonatal immunometabolic responses to infection, vaccination and development of chronic disease, and potentially developing interventions to perturb implicated pathways for clinical benefit.

Below are my comments to improve this important manuscript

1. The dose of a microbial stimulus can influence the direction of the immune response in various ways. For example, it is well-known that high doses of a microbial stimulus may result in immune exhaustion or paralysis, while lower doses may result in a pro-inflammatory phenotype. According to the Methods, the authors used variable doses of LPS for different experiments. It would be helpful for the reader if the authors **a)** indicated the dose and duration of LPS treatment in each figure legend, and **b)** provided justification as to why they chose different LPS doses. I would be keen to see if there are age-specific responses using the same dose of LPS across experiments.

Answer: We thank the reviewer for the hint and fully agree that the description of experimental settings needs better elaboration to help the reader. In the Methods section as well as in the figure legends, we now clearly indicate the doses and durations used for LPS treatments in dependence of the cell type (M₀, MDM, MNC) and intended model and read-out.

Isolated monocytes have been activated for 4h with 10 ng/ml LPS in gene expression studies and for 16h with 100 ng/ml LPS in cytokine studies.

In pretreatment experiments, isolated monocytes were low-dose pre-treated for 16h with 0.1 ng/ml LPS followed by an activation with 1 µg/ml LPS for 4h. We included a justification of dose and duration chosen for the pretreatment conditions in the Methods section (please see also Reviewer #1/point 6).

MDM were activated for 4h and 16h with 10 ng/ml LPS since human monocytes acquire increased sensitivity to LPS during differentiation to macrophages (Vishnyakova et al., 2021, PMID: 33930675; Jungi et al., 1994, PMID: 8125167).

Metabolic pathway activities were determined by flow cytometry-based SCENITH studies in MNCs. In pretests in AD MNCs, we found that the activation of human MNCs with 1 µg/ml of LPS achieved the most differential LPS-induced changes in metabolic pathway activities of M₀.

2. The terms metabolic reprogramming and immune training are often used in the text to describe some phenomena observed by the authors that do not necessarily reflect these processes, as defined in the literature, also cited in the references of this manuscript. Innate immune training is characterized by long-term epigenetic and metabolic reprogramming of innate immune cells, which manifest days to weeks after an initial stimulation and are induced by a secondary unrelated stimulus after the cell has returned to its pre-stimulated state. This is different from the acute changes observed shortly after LPS stimulation as performed in this paper.

Answer: We fully agree with the reviewer that trained immunity is driven by long-term changes in gene expression and metabolism of innate immune cells. The concept of innate trained immunity suggests that innate immune cells can develop a form of memory after exposure to certain stimuli, leading to enhanced responses upon subsequent encounters with similar or different pathogens. Trained immunity does not last as long as adaptive immunity, but it is still longer-lasting than the immediate responses of classical innate immunity.

We agree that the time between the primary and secondary stimuli is short and the stimuli are the same. However, modelling trained immunity in monocytes is tricky. *In vivo* innate training has been demonstrated in murine and a few human studies by priming the individual and assessing MNCs or monocytes at later time points (Quintin et al., 2012, PMID: 22901542; Arts, Carvalho et al., 2016; Zhang et al., 2022, PMID: 35133977). However, *ex vivo*, most studies used primed monocytes and only reassessed them after differentiation into MDM after 6 days of cultivation (Quintin et al., 2012, PMID: 22901542; Cheng et al., 2014; Arts, Carvalho et al., 2016; Bekkering et al., 2016, PMID: 24903093; Zhang et al., 2022, PMID: 35133977). Reassessing monocytes (which was our explicit aim) is impossible after extended culture times due to their restricted *ex vivo* lifespan (24h-48h; please see also our answer to reviewer #1/point 5 and 6).

We acknowledge the reviewer's concern that using the term "immune training" or "microbial training" is imprecise as we capture in fact the immunometabolic effects of a LPS pretreatment. We followed the recommendation of reviewer #1 and replaced "microbial training/education" by "LPS-pretreatment" throughout the revised manuscript.

Using LPS for priming as well as restimulation was attractive for us as it allowed demonstrating that LPS tolerance was only induced in AD monocytes but not in NEO monocytes where LPS pretreatment oppositely induced priming. This suggested that the outcome "Priming" vs "Tolerance" is dependent on the initial metabolic programming at primary stimulus exposure, which is supported by the new experimental data shown in Fig. 3f-i (see our answer to point 8-b).

3. lines 211-213: The authors employ an interesting experimental design based on a ketogenic diet intervention to determine if dietary factors can explain immunometabolic differences between NEO and AD. However, the comparison of breast-milk to a ketogenic diet is inaccurate. Approximately 39% of breastmilk consists of carbs which makes it a moderately high carb content diet. Also, breastmilk composition changes rapidly during the first few months of life, which is when major immune maturation takes place. In addition, ketosis in newborns does not occur due to their diet, but is physiologic due to their developmental needs: ketones are a precursor to important metabolites that help with brain development, including cholesterol and amino acids.

Answer: The reviewer is right in objecting that our statement of breast milk being a sort of ketogenic diet is inaccurate. The reviewer is also fully right that ketosis in newborns does not occur due to their diet, but is physiological due to their developmental needs. We reworded the introductory part of this section profoundly and also the abstract. Thereby, we hope to better explain our rationale underlying these experiments. Please, see our answers to reviewer #1/points 8 and 9.

4. lines 239-240: There were higher rates of NEO Mo death compared to AD. According to the Methods, those cultures were supplemented with 10% heat-inactivated human AD blood plasma, implying that no age-specific plasma was used for each age group. There is plenty of literature emphasizing the importance of using age-specific plasma in human in vitro assays (Alipour, R., Fatemi, A., Alsahebhosul, F. et al. Autologous plasma versus fetal calf serum as a supplement for the culture of neutrophils. *BMC Res Notes* 13, 39 (2020). <https://doi.org/10.1186/s13104-020-4902-z>; Angelidou A, et al. Human Newborn Monocytes Demonstrate Distinct BCG-Induced Primary and Trained Innate Cytokine Production and Metabolic Activation In Vitro. *Front Immunol.* 2021 Jul 13;12:674334. doi: 10.3389/fimmu.2021.674334.), especially since neonatal plasma is particularly rich in age-specific immunomodulatory factors (Sanchez-Schmitz, G. & Levy, O. Development of newborn and infant vaccines. *Sci. Transl. Med.* 3, 9027. <https://doi.org/10.1126/scitranslmed.3001880> (2011).) and other microphysiologic components that can influence cellular well-being, immune responsiveness and viability.

Answer: The reviewer suggests that using age-specific plasma for MDM differentiation would have been important.

We were specifically interested in the cell-intrinsic immunometabolic programming of neonatal and adult monocytes and related capacity to differentiate into MDM. Varying both parameters, cells and plasma, at once makes it difficult ascribing differential outcomes to either the cellular programming or the plasma composition. It was therefore important for us to use the same standardized and well-defined culture medium condition. In case of MDM differentiation, we chose using a plasma pool of healthy adult blood donors which in terms of compositional variability is more consistent than autologous plasma from different donors, especially from cord blood donors. Plasma derived from cord blood tends to vary considerably, particularly in proteins, immunoglobulins, and clotting factors, depending on maternal-, pregnancy-, delivery- and child-related factors (e.g. Van Pee et al., 2024, PMID: 38743419; Pirr et al., 2019, PMID: 33931974; Shokry et al., 2019, PMID: 30725264; Austermann et al., 2014; Strunk et al., 2009, PMID: 19383861).

However, the reviewer is right that age-specific plasma components might impact the immunometabolic programming of monocytes and might provide further guidance in explaining the observed differences. Therefore, we performed new experiments using blood plasma from healthy adult donors (AD-PL) and healthy newborn donors (NEO-PL from cord blood) in an autologous and cross-heterologous manner for the differentiation AD and NEO Mo into MDM (new Supplementary Fig. 6d-f). We observed no significant differences between AD-PL and NEO-PL in terms of total cell yield (Supplementary Fig. 6d), apoptosis rates (Supplementary Fig. 6e) and the proportions of CD68⁺ MDMs (Supplementary Fig. 6f). However, the differences between NEO MDM and AD MDM differentiated with NEO-PL showed higher heterogeneity within age-groups and the differences between NEO and AD MDM were less clear, likely due to the higher compositional variability of cord blood. To better understand the complexity of

whether and how specific plasma components potentially impact the immunometabolic programming of immune cells we plan to perform proteomic profiling of a high number of plasma samples in an ongoing human birth cohort study (Hartmann et al., 2024, PMID: 39355253) that will be integrated with corresponding cell metabolic studies in monocytes.

5. Are there any paired samples between experiments or were all assays run in different datasets? Given the wide inter-donor variability in the readouts tested, running more than 1 assay in the same donor can strengthen the credibility of the results, especially for experiments where the N is low.

Answer: Unfortunately, the amount of cord blood and even more of peripheral infant blood obtainable from newborns and infants is restricted compared to that of adult blood. This limits the number of experiments that can be performed at once, especially, when working with monocytes isolated from the blood samples. Therefore, RNAseq, SCENITH, isotope tracing, and MDM differentiation studies represent separate experimental series done with unrelated samples. However, we actually consider it rather as a strength of the study that data from independent experiments and donors with different read-outs contributed coherently to the same concept.

6. Figure 1:

panels e and f: I guess the results plotted reflect the LPS-treated conditions. Can the authors also plot the LPS-untreated controls? For panels h-l, it is unclear if the results plotted reflect the LPS-stimulated conditions or quiescent cellular states. Please clarify.

Answer: We added the LPS-untreated controls to Fig. 1e,f. After 4h of cultivation, there was no significant cytokine accumulation detectable in both NEO and AD monocyte cultures.

The isotope tracing shown in panels h-l were done in LPS-untreated cells. We added "Untreated" to the figure legend for clarification.

7. Figure 2:

a) SCENITH: were these studies performed on MNCs or Mo? Please be precise (figure legend states Mo, panel a states MNC, methods state MNC). MNC includes cell populations other than Mo.

Answer: We apologize for the confusion and reworded the figure legend. SCENITH was done in MNCs with metabolic parameters determined and plotted for the monocyte population.

b) panel g-j: why not plot all age groups? Can the authors use different shapes to show which data points belong to each age group? Alternatively, can the authors plot the cytokine data for each age group?

Answer: Only cord blood and adult blood allowed isolating enough MNCs and monocytes to perform SCENITH and LPS stimulation assays at once. As suggested by the reviewer, we now use different shapes in the plots of Fig. 2g-j to show which data point belongs to each age group.

c) panel k: how were the clinical data variables chosen? The number of vaccinations or vaccines does not apply to the newborn group, who presumably is not yet vaccinated (especially if cord blood was used). Also, it is a huge leap to assume that high number of

siblings is associated with metabolic reprogramming because of higher microbial exposure. Also, the effect size for this observation is really small.

Answer: In our study, the selection of clinical variables for inclusion in the generalized linear model was guided by their hypothesized relevance to metabolic reprogramming during early life, based on both biological plausibility and prior research (as cited below). We used a Poisson GLM with a log-link function to model the count-based nature of our metabolic activity measures, ensuring that the predictors' effects were multiplicative on the expected outcomes. This approach is crucial for accurately reflecting the exponential relationships typical of biological systems. Thereby, age, sex, gestational age (GA), mode of delivery (MOD), body weight, number of vaccinations, number of vaccines, and number of siblings were chosen as predictors due to their potential roles in influencing the metabolic phenotypes of monocytes.

Age was the primary covariate due to the observed age-dependent shifts in energy pathways.

The association between gestational age and cell metabolism is a significant area of research. Generally, the GA is well known to have an influence on metabolism and cell metabolism, particularly when comparing preterm and term infants (Li et al., 2020, PMID: 31894879; Kan et al., 2018; Hovi et al., 2007, PMID: 17507704; Singhal & Lucas, 2004, PMID: 15145640). Admittedly, we did not expect a huge impact of the GA on the cell metabolisms in our specific model as this covariate varied only slightly in our study population due to the inclusion criterium of all children having been born at term. Nevertheless, we found it worth including considering prior research.

Sex has been chosen since it has profound effects on cell metabolism, influencing everything from basic metabolic rate and energy expenditure to the risk of developing metabolic disorders. Biological differences such as sex chromosomes and hormones shape how metabolism functions across different stages of life (Wiese et al., 2023, PMID: 37598068; Dearden et al. 2018, PMID: 29773464).

Body weight and metabolism are closely interconnected, influencing each other in various ways (Rosenbaum & Leibel, 2010, PMID: 20935667; Hall & Kahan, 2018, PMID: 29156185). Likewise, body weight and cell metabolism are closely linked, as the metabolic processes within cells play a crucial role in determining overall body weight (Hotamisligil, 2006, PMID: 17167474; Houten & Auwerx, 2004, PMID: 15454076).

The MOD, number of vaccinations and vaccines, and number of siblings were included as they were the most reliably documented variables that are directly or indirectly associated with microbial exposures that were potentially differential among the study participants. As pointed out in the manuscript, infections were unfortunately insufficiently documented and validated to allow reliable data integration.

The gut microbiota is our most important source of commensal microbial exposure. Its composition is well known to differ considerably in dependence of the MOD (Reyman et al., 2019, PMID: 31676793; Dominguez-Bello et al., 2010, PMID: 20566857).

Though the number of vaccinations and vaccines did not significantly vary in our study population, we found them important to include into the model. Though vaccination is certainly not relevant in newborns, including "NEO age" as categorical variable in the model is important to control for confounding effects in older groups, consistent with the principle of over-parameterization in regression analysis. This strategy helps to ensure the precise estimation of the main effects, particularly age. Creating categorical variables is the crucial step that allows modelling each age group separately. In complex models, the additional inclusion of

interaction terms would allow to determine how the relationship between the covariate and the outcome changes across different age groups. However, as the reviewer rightly notes, these relationships like those between the vaccination variables and NEO age are obvious and plausible in our model, not requiring further computation.

The sibling effect on microbial exposure plays a significant role in shaping the immune system and overall health. Siblings contribute to a more diverse microbial environment, which can help in the development of a robust immune system, potentially reducing the risk of allergic and autoimmune diseases (Gao et al., 2023, PMID: 37150361; Tamburini et al., 2016, PMID: 27387886; Laursen et al., 2015, PMID: 26231752; Benn et al., 2004, PMID: 15121716; Ball et al., 2000, PMID: 10954761; Strachan, 1989, PMID: 2513902). Therefore, based on the hypothesis of a higher microbial turnover in sibling-rich families, we included the number of siblings into our model assuming a sibling effect on the cellular immunometabolism. The observed effect size was small but statistically significant, reflecting a modest yet real biological impact. In our Poisson model, this small effect size is interpreted as the log of the rate ratio, indicating a proportional change in OXPPOS activity with each additional sibling. Our model used standardized estimates adjusted with the Benjamini-Hochberg procedure to control for false discoveries, providing robust p-values despite multiple comparisons.

We acknowledge the reviewer's concerns about the applicability of vaccination variables and the speculative nature of the siblings' effect. We added the above-mentioned citations in the respective Results section to corroborate our assumption that a high number of siblings is associated with a high microbial turnover in families. Furthermore, we refined our discussion to emphasize that the sibling effect, while significant, is small and should be viewed as preliminary as follows: *"We found a small though significant decreasing sibling effect on the OXPPOS activity, preliminarily suggesting a link to a more diverse microbial exposure¹²⁻¹⁶."*

8. Figure 3: Microbial education induces a metabolic shift towards an AD-like phenotype:

a) panel a+b: I am not sure what the authors mean by microbial "education". What they performed is microbial stimulation with LPS (please clarify dose). The relevant comparisons here are LPS vs control, not LPS 4h vs 24h, which indicates the kinetics of the response [unless I misunderstood the design and the LPS 4h and 24h refer to pretreatment for 4h and re-stimulation for 24h]. My interpretation of the data shown is that in NEO Mo, LPS stimulation decreased mitochondrial dependence over the control (baseline) but had no effect on glycolytic dependence (since change from baseline was not significant). In the AD Mo, LPS stimulation had no effect on mitochondrial dependence compared to control, but significantly increased glycolytic dependence (though the effect size seems quite small). Therefore, I find the summary in page 9, line 191-194 inaccurate. Since NEO and AD have different baseline metabolic needs, comparing NEO and AD (such as is done in Suppl figure 3) is meaningful only if their baseline is somehow corrected for/accounted for.

Answer: The reviewer is right that the term "microbial education" is misleading and imprecise wherefore we abandoned using this term and refined the wording by indicating clearly where we performed LPS treatment and pretreatment (see also our answer to point 2). The reviewer is right that Fig. 3a-c show SCENITH studies comparing untreated control, 4h and 24h LPS treated settings (dose indicated in the Methods section and now also in the figure legend). The reviewer rightly noticed that LPS stimulation had a clear and higher impact on the mitochondrial dependence than the glycolytic dependence of NEO Mo, which only slightly but not significantly increased compared to baseline. To not give the wrong impression and to make this point

clearer we have changed the wording related to Fig. 3b (now: “*LPS stimulation increased glycolysis in NEO Mo only slightly with strong inter-individual variance but in AD Mo uniformly and highly significant (Fig. 3b).*”) and moderated the summary (now: “*Summarized, these studies suggested that a short-term single LPS treatment induced a metabolic reprogramming of NEO Mo towards an AD-like baseline phenotype, particularly by decreasing OXPHOS and slightly increasing glycolysis.*”).

b) panels c-e: These experiments were reportedly done to test whether the LPS-induced metabolic shift in NEO Mos also comes with a functional maturation towards an “AD-like immune phenotype”. What is the hypothesized AD functional phenotype and in which state are the authors referring to (LPS-treated, LPS-pretreated)? Why is pre-treatment with LPS required to show changes in the functional maturation of Mo? Flow cytometry for cellular differentiation/maturation markers in LPS-treated Mo would have been more informative. These panels are distracting and the message is unclear. It would be more helpful to show cytokine/chemokine concentrations after primary and secondary stimulation to determine changes compared to the immediately pre-existing cellular state. In lines 197-202, the authors report that “LPS pre-treatment followed by LPS restimulation promoted a stronger induction of TNF α in NEO Mo corroborating our finding that LPS exposure promotes glycolytic activity on NEO Mos”. Do they have metabolic data on the same samples that were pre-treated and restimulated with LPS? It is very likely that pre-treatment and re-stimulation could influence metabolic shifts differently than a single microbial stimulation. If those experiments were not done, this claim cannot be made (also, as mentioned above, there was non-significant effect on NEO glycolysis in panel b). One way to address this question would be to block glycolysis and measure the effects on cytokine secretion.

Answer: We thank the reviewer for the constructive advice to perform metabolic blocking experiments, which we agree are crucial to support our concept and improve the manuscript.

Regarding the AD functional phenotype, we are referring to the primary response of AD Mo to LPS which NEO Mo only achieve after LPS pretreatment.

Low-dose LPS pretreatment was done to allow long-term LPS exposure. Low-dose LPS was chosen to ensure well-being of monocytes in the frame of their *ex vivo* lifespan (see above) and avoid exceeding activation induced cell death (AICD).

We found functional studies in LPS-treated Mo more informative than phenotyping by flow cytometry as we were primarily interested in how metabolism and cell functions are linked. We actually show the cytokine/chemokine concentrations after primary and secondary stimulation in Fig. 3c-e to demonstrate the changes after long-term LPS exposure and the immediately pre-existing states before primary and secondary LPS stimulation.

To corroborate our statement that long-term LPS exposure of NEO Mo promotes their glycolytic activity, we performed new experiments assessing the effects of blocking glycolysis and OXPHOS on the LPS response of not pretreated (primary LPS response) and LPS-pretreated (secondary LPS response) NEO and AD Mo. The description of the experimental details has been added to the Methods section. The statement in former lines 197-202 has been mitigated as follows: “[...] *suggesting that continuous low-dose LPS exposure promotes the glycolytic activity of NEO Mo.*”. The results of the new experiments are now illustrated in the new Fig. 3f-i and described at the end of the third Results section as follows: “*Metabolic blocking experiments corroborated our assumption that a strong inflammatory response of AD Mo to a primary LPS stimulus and NEO Mo to a secondary LPS stimulus after LPS priming is linked to their glycolytic activity. In line with the low glycolytic activity of untreated NEO Mo,*

blocking glycolysis suppressed their TNF- α response but to a lesser extent and not significantly as in untreated AD Mo (Fig. 3f). In contrast, after LPS pretreatment, the then enhanced TNF- α response of NEO Mo to secondary LPS activation was significantly inhibited when glycolysis was blocked, while no effect was detectable in LPS-pretreated and thus tolerized AD Mo (Fig. 3g). Blocking OXPHOS had no clear effects on TNF- α responses neither on the primary or secondary responses of NEO Mo nor that of AD Mo (Fig. 3h,i)).”.

Reviewer #3:

In the manuscript entitled “Oxidative phosphorylation is a key feature of neonatal monocyte immunometabolism promoting myeloid differentiation after birth” Ehlers and coworkers combine transcriptomic, metabolic, and immunological approaches in monocytes from healthy individuals at different ages to unravel the ontogenetic differences in the metabolic features and their relationship with myeloid differentiation and inflammatory response. In general this study is very relevant to the field, as it contributes to the characterization of human macrophages and their metabolic features with respect to age and, as such, represents an important step toward the ontogenetic knowledge of human macrophages metabolism and function, which is currently quite limited if compared to murine macrophages. For this reason I believe that this work fills a gap in our knowledge of human macrophages. In particular, the authors are presenting a cutting edge approach to unravel age-dependent differences in human monocytes and macrophages.

We thank the reviewer for the encouraging words and for appreciating that we try to fill a knowledge gap in understanding age-dependent immunometabolism as a meaningful physiological program.

However considering the methods applied in this study, I believe that this manuscript would fit much better a more specific system biology journal.

Answer: As the reviewer rightly states, the study is very relevant to the field for which reason we aim to reach out to a broad readership by submitting it to *Nature Communications* instead of a specific system biology journal. In fact, so far, all previous studies with a specific focus on neonatal cell metabolism have been published by *Nature Communications*. (Kan et al., 2018; Dreschers et al., 2019). Our results give important further insights into the ontogeny of innate cell immunometabolism. They shed new light on the physiological meaning of how neonatal Mo are programmed and to what biological processes their metabolic programming is linked. We believe that this is of interest to a broad readership.

The conclusions on the dependency of NEO M0 on OXPHOS are not validated and demonstrated only limitedly at a molecular level.

Answer: We thank the reviewer for raising concerns that our conclusions on the dependency of NEO Mo on OXPHOS are not validated and demonstrated. We addressed these concerns as follows. As shown in Fig. 5i-p, blocking OXPHOS impaired the quality of MDM differentiation, particularly when derived from NEO Mo that are characterized by high OXPHOS activity. New metabolic blocking experiments in monocytes (new Fig. 3f-i) now confirm that their glycolytic activity but not their OXPHOS activity determines their inflammatory responsiveness (see also our answer to reviewer #2/point 8-b). Population-based HUVA studies corroborated the association of OXPHOS with gene programs involved in myeloid differentiation (Fig. 7b-n). New TF inhibition experiments shown in a new Fig. 8 further support the link between OXPHOS and myeloid differentiation as the inhibition TFs that promote OXPHOS (E2F1 and MYB) (Fig. 8a) strongly impede high-quality myeloid differentiation (Fig. 8e).

The mechanisms underlying the relationship between the transcriptional programs being activated in NEO vs AD macrophages and the corresponding metabolic outcomes are not demonstrated. This might require to incorporate a wide amount of additional new data.

Answer: We thank the reviewer for pointing out that the relationship between the transcriptional programs and metabolic outcomes should be demonstrated experimentally. Given the challenge of manipulating primary human monocytes without heavily distorting their primary transcriptional and metabolic programming required a considerable number of experimental tests and establishments to address to the reviewer's request. In the revised manuscript, we now incorporated a wide amount of additional new data delineating what consequences the inhibition of our candidate transcription factors has on cell metabolism and MDM differentiation. To assess the effect of inhibition of the candidate TFs on predicted functions, we avoided genetic editing of Mo that distorts their primary transcriptional and metabolic programming and decided to employ pharmacological inhibitors:

- HLM006474, an inhibitor of E2F1 (Ma et al., 2008, PMID: 18676853; Shirakawa et al., 2022, PMID: 36198264),
- All-trans retinoic acid (ATRA), an inhibitor of MYB (Mandelbaum et al., 2018, PMID: 30209067; Hanna et al., 2021, PMID: 34091189),
- Fludarabine (FA) an inhibitor of STAT1 (Frank et al., 1999, PMID: 10202937; Jiang et al., 2024, PMID: 38509096), and
- Camptothecin (CPT) as inhibitor of FLI1 (Wang et al., 2021, PMID: 33559345; Schutt et al., 2022, PMID: 36074578).

The new data have been illustrated in the new Fig. 8 and reported in a new last Results paragraph. Even though these inhibitors might have broader effects than exclusively inhibiting E2F1, MYB, STAT1, and FLI1, respectively, the consequences for metabolic pathways activities and MDM differentiation largely supported the computationally predicted TF functions. The inhibition of E2F1 and MYB decreased the OXPHOS of monocytes and impaired their differentiation into MDMs (Fig. 8a,e), which is in line with the phenotype of NEO Mo that express E2F1 and MYB at significantly higher levels than AD Mo (Fig. 7a). In contrast, the inhibition of STAT1, FLI1 and also MYB strongly promoted cell death during myeloid differentiation (Fig. 8c,d), which was however unlikely linked to glycolysis since glycolysis was inhibited by MYB but promoted by FLI1 (Fig. 8b). The data strongly support that high E2F1 and MYB expression in NEO Mo promote OXPHOS and cell differentiation that shifts to low E2F1 and MYB expression and high STAT1 expression with increasing age, driving glycolysis and inflammatory activity (see also new Fig. 3f,g) in AD Mo in liaison with FLI1.

Point-by-point response to Reviewer #2:

1. Supplementary figure 6 shows panels of cell numbers and percentages of apoptotic and CD68+ cells. The data show that (high) non-physiologic concentrations of growth factors promote cellular growth and differentiation compared to physiologic concentrations of growth factors contained in human plasma. The authors also show new culture data using AD blood plasma for AD cells and NEO blood plasma for NEO cells (unclear if NEO blood plasma is pooled or not), and show better results when using same-age human plasma (NEO_NEO-PL and AD_AD-PL) vs different age (NEO_AD-PL and AD_NEO-PL) human plasma, meaning better cell numbers (d, better though not significantly different by age), lower % of apoptotic cells (e, significant for NEO only) and comparable number of CD68+ cells (f). Stats between NEO_NEO-PL and AD_AD-PL are missing, but in my opinion, using plasma from the same age group (pooled, if not from the same donor) is the gold standard, especially if the authors are trying to show relevance of their ex vivo model for in vivo human biology. Differences between same age vs different age plasma in their effects on cellular differentiation and effector functions (vs just cell numbers) are not shown.

In their rebuttal the authors state that they were “specifically interested in the cell-intrinsic immunometabolic programming of neonatal and adult monocytes and related capacity to differentiate into MDM. Varying both parameters, cells and plasma, at once makes it difficult ascribing differential outcomes to either the cellular programming or the plasma composition. It was therefore important for us to use the same standardized and well-defined culture medium condition”. It is well known that the cellular microenvironment (in this case the culture medium and any added factors) affects “cell-intrinsic” responses to a great degree. The peculiarities of neonatal immune responses are largely attributable to the unique composition of their plasma and dynamic changes thereof, which is very different from that of adults. The use of plasma from a different age group could obscure soluble plasma-based ontogenic differences that shape cellular immune responses (Pettengill et al, Soluble mediators regulating immunity in early life, Front Immunol 2014). Although the authors express concern about varying 2 parameters (cells and plasma), I would argue that those 2 parameters are interdependent and should be paired by age (NEO cells in NEO plasma and AD cells in AD plasma), especially if the work describes age-specific effects on metabolism. This is a major limitation in the model that should be highlighted in the discussion.

Answer: We appreciate the reviewer’s concern about our model of myeloid differentiation. We fully agree that the age-specificity of the plasma composition is very important to consider when assessing blood immune cells (as shown in our previous work in NEO/AD monocytes (Ulas et al. 2017)). However, modelling myeloid differentiation *in vitro* can reliably only assess cell-intrinsic signatures (under standardized medium conditions) as macrophage differentiation *in vivo* takes place in tissues. It remains questionable whether adult blood plasma and cord blood plasma are appropriate to model age-specific tissue microenvironments.

Cord blood plasma is a very special product with its composition being highly dependent on maternal and birth-related factors and changing dramatically within hours to days after birth. Our own experiences are (and as demonstrated here) that cord blood plasma increases the variability of many cell differentiation assays considerably. Of note, though rich in growth factors, professional tissue engineers do not use cord blood plasma as it is quite undefined and can also come with known/unknown inhibitory factors.

To notify the readership about these reflections we added the following paragraph to the Discussion: “*In this work, we were interested in the age-specific cell-intrinsic myeloid*

*differentiation capacity and therefore used standardized culture conditions with pooled AD blood plasma. This approach neglects potential age-specific influences from the tissue microenvironment on macrophage differentiation in vivo. Using blood plasma is presumably generally not appropriate to model tissue microenvironments, particularly not that of neonates when using cord blood plasma given its dramatic compositional changes within the first days after birth*⁵⁰. Nevertheless, we tested the effect of cord blood plasma and observed that MDM differentiation tended to be less supported than by AD blood plasma and lost significance regarding differences between NEO and AD MDM. We ascribe this to the high compositional variability of cord blood plasma^{25-27,50} that can be rich in factors promoting (e.g. GM-CSF) as well as regulating myeloid differentiation (e.g. TGF- β)^{51,52}.”. We leave it to the editor’s decision whether this addition exceeds the length and the scope of this manuscript.

The statistics of NEO_NEO-PL and AD_AD-PL have been added to Supplementary Fig.6d-f.

The cord blood plasma has been pooled. We added this information to the Method section.

2. #6: In response to authors’ response:… We added the LPS-untreated controls to Fig. 1e,f. After 4h of cultivation, there was no significant cytokine accumulation detectable in both NEO and AD monocyte cultures.

From what I can see and what is noted in the text, there IS significant cytokine accumulation in both NEO and AD monocyte cultures: “Higher LPS-induced TNF-a production by AD Mo compared to NEO could also be validated at the protein level (Fig. 1e)” and “NEO Mo distinguish from AD Mo by a higher GM-CSF production (Fig. 1f)”.

Answer: There is a misunderstanding here. Our comment of “no significant cytokine accumulation” referred to the LPS-untreated NEO and AD controls.

3. #8: In response to authors’ response: …we have changed the wording related to Fig. 3b (now: “LPS stimulation increased glycolysis in NEO Mo only slightly with strong inter-individual variance but in AD Mo uniformly and highly significant (Fig. 3b).”) and moderated the summary (now: “Summarized, these studies suggested that a short-term single LPS treatment induced a metabolic reprogramming of NEO Mo towards an AD-like baseline phenotype, particularly by decreasing OXPHOS and slightly increasing glycolysis.”).

I believe Fig 3a-b have tested MNCs not Mos. If so, please correct.

Answer: The SCENITH assay uses MNCs but since it is a FACS-based method it allows determining the metabolic parameters for specific cell population depending on the gating, in this case monocytes. No correction necessary.

4. Figure 2 g-j: consider different regression lines for NEO and AD as the trends appear to differ by age.

Answer: We tested this. The trends of the regression lines do not fundamentally differ by age but became underpowered (Figure R1 for the reviewer). As we found these plots not really helpful and therefore decided not to replace the existing ones.

Fig. R1. Correlations between LPS-induced cytokine production dependence on glycolysis and OXPHOS at baseline. Correlations between the LPS-induced production of TNF- α (a,c) and GM-CSF (b,d) by NEO (pink) and AD (dark grey) Mo and their baseline glycolytic (a,b) and OXPHOS (c,d) dependence. Indicated are best fit regression lines for the correlations in NEO, AD and both (ALL) Mo. Pearson's correlation coefficients (r) and p values of correlations are indicated for ALL.

5. Lines 456-459: "In our study, LPS pretreatment had age-dependently opposite effects and enhanced subsequent LPS responses in NEO Mo but paralyzed those of AD Mo, suggesting that the initial state of metabolic programming might determine the educational outcome." The use of the phrase "educational outcome" hints at trained immunity. Although this work may have implications for innate immune training, the authors did not model monocytic trained immunity here, as they only cultured Mos for 24-48h. Please rephrase.

Answer: We rephrased as follows: "[...], suggesting that the initial state of metabolic programming might determine the outcome of LPS pretreatment".